# Swap Agnostic Learning,
# or Characterizing Omniprediction via Multicalibration

**Parikshit Gopalan**
Apple, Inc.

**Michael P. Kim**
UC Berkeley

**Omer Reingold**
Stanford University

## Abstract

We introduce and study Swap Agnostic Learning. The problem can be phrased as a game between a *predictor* and an *adversary*: first, the predictor selects a hypothesis $h$; then, the adversary plays in response, and for each level set of the predictor $\{x \in \mathcal{X} : h(x) = v\}$ selects a loss-minimizing hypothesis $c_v \in \mathcal{C}$; the predictor wins if $p$ competes with the adaptive adversary's loss. Despite the strength of the adversary, our main result demonstrates the feasibility Swap Agnostic Learning for any convex loss. Somewhat surprisingly, the result follows by proving an *equivalence* between Swap Agnostic Learning and swap variants of the recent notions Omniprediction [15] and Multicalibration [20]. Beyond this equivalence, we establish further connections to the literature on Outcome Indistinguishability [6, 14], revealing a unified notion of OI that captures all existing notions of omniprediction and multicalibration.

## 1 Introduction

Since its inception as an extension to Valiant's PAC framework [30, 19, 23], Agnostic Learning has been the central problem of supervised learning theory. Agnostic learning frames the task of supervised learning through loss minimization: given a loss function $\ell$, a hypothesis class $\mathcal{C}$, and $\varepsilon \geq 0$, a predictor $h$ is an agnostic learner if it achieves loss that competes with the minimal achievable within the hypothesis class.

$$\mathbf{E}[\ell(\mathbf{y}, h(\mathbf{x}))] \leq \min_{c \in \mathcal{C}} \mathbf{E}[\ell(\mathbf{y}, c(\mathbf{x}))] + \varepsilon \tag{1}$$

While the agnostic learning paradigm has been remarkably successful, in recent years, researchers have investigated alternative learning paradigms to address concerns of the modern prediction pipeline, including fairness and robustness. The work of [20] introduced *multicalibration* as a new paradigm for learning fair predictors. Multicalibration asserts fairness as a first-order goal, requiring that predictions appear calibrated even conditioned on membership in one of a potentially-huge collection of subgroups $\mathcal{C}$ of the domain.[1] As a solution concept, multicalibration can be achieved efficiently using a weak learner for $\mathcal{C}$ to identify subgroups where predictions are miscalibrated. In contrast to agnostic learning, multicalibration does not make reference to minimizing any loss.

Yet, it turns out that there are surprising connections between multicalibration and agnostic learning. This connection was first discovered in the work of [15], who introduced the notion of *omniprediction* as a new solution concept in supervised learning. Intuitively, an omnipredictor is a single predictor that provides the agnostic learning guarantee for many losses simultaneously. More formally, for a collection of loss functions $\mathcal{L}$ and a hypothesis class $\mathcal{C}$, a predictor is an omnipredictor if *for any loss in the collection* $\ell \in \mathcal{L}$, the predictions achieve loss that competes with the minimal achievable

---

[1]This collection of subgroups is suggestively denoted by $\mathcal{C}$, indicating that (in correspondence with a hypothesis class) subgroup membership can be computed using models of bounded capacity, such as small decision trees, halfspaces, or neural networks.

37th Conference on Neural Information Processing Systems (NeurIPS 2023).

within the hypothesis class.[2] The main result of [15] demonstrates that multicalibrated predictors are omnipredictors for $\mathcal{L}_{\mathrm{cvx}}$ the class of all convex loss functions. In other words, the multicalibration framework is capable of guaranteeing agnostic learning in a very strong sense: learning a single multicalibrated predictor gives an agnostic learner, for every target convex loss $\ell \in \mathcal{L}_{\mathrm{cvx}}$. The results of [15] stand in contrast to the convential wisdom that optimizing predictions for different loss functions requires a separate training procedure for each loss.

On the surface, multicalibration seems like a fundamentally different approach to supervised learning than agnostic learning. The implication of omniprediction from multicalibration, however, suggests a deeper connection between the notions. In fact, follow-up work of [14] gave new constructions of omnipredictors (for different loss classes $\mathcal{L}$), similarly deriving the guarantees from variants of multicalibration. To summarize the state of the art, we have constructions of omnipredictors of various flavors, and all such constructions rely on some variant of multicalibration. While multicalibration suffices to guarantee omniprediction, a glaring question remains in the development of this theory: *Is multicalibration necessary for omniprediction?* We investigate this question, exploring the connections between agnostic learning, notions of omniprediction, and multicalibration.

**Our Contributions.**    In this work, we provide a new perspective that unifies these seemingly-incomparable paradigms for supervised learning. Key to this perspective, we introduce a new learning task, which we call Swap Agnostic Learning. This *swap* variant of agnostic learning is inspired by the notion of swap regret in the online learning literature [10, 2], where the learner must achieve vanishing regret, not simply overall, but even conditioned on their decisions. While swap agnostic learning is a natural extension, at first glance, it is a considerably stronger goal than standard agnostic learning. Nevertheless, our work demonstrates an efficient algorithm for swap agnostic learning for any convex loss function, leveraging only a weak agnostic learner for the hypothesis class. The algorithm follows by discovering a surprising connection: swap agnostic learning and swap variants of omniprediction and multicalibration are actually equivalent. In other words, once we move to swap variants, multicalibration *is* necessary for omniprediction, and even for agnostic learning. We show how the original multicalibration algorithm of [20] actually guarantees the stronger goal of "swap multicalibration" (and thus, swap omniprediction and swap agnostic learning). Our results provide an exact characterization of these swap learning notions, as well as relationships between other learning desiderata explored in recent works [6, 14].

Our motivation for introducing Swap Agnostic Learning comes from trying to understand the relationship between (standard) multicalibration and omniprediction. Prior work shows that multicalibration implies (convex) omniprediction, but leaves open the question of whether omniprediction implies multicalibration. The present work sheds new light on this question, suggesting that the asnwer is no: standard omniprediction does not imply multicalibration. The answer to this question comes by introducing the idea of Swap Learning, in the agnostic learning/omniprediction setting, as well as in multicalibration. Importantly, for multicalibration, Swap multicalibration and Standard multicalibration are essentially the same notion. Claim 2.9, as well as our analysis of prior algorithms for multicalibration, shows that standard multicalibration is already strong enough to capture the swap variant. The distinction (definitionally and algorithmically) between swap and standard multicalibration is minimal: our results suggest that one should really think of them as a single notion of multicalibration. For agnostic learning and omniprediction, however, Swap Learning is much stronger than Standard Learning. We prove separations between swap and standard omniprediction in Appendix B. The separations are summarized in figure 1. This tells us that (swap) multicalibration is equivalent to the stronger notion of swap omniprediction, which is provably stronger than standard omnipreidiction.

**Organization.**    We continue the manuscript with formal setup and defintions of Swap Agnostic Learning, Swap Omniprediction, and Swap Multicalibration. Then, in our main result, we prove the equivalence of these notions. This equivalence suggests an efficient algorithm for Swap Agnostic Learning. We conclude with an overview of our other results, as well as related work and discussion. Throughout the manuscript, we include formal definitions of key notions and proofs of essential claims. We introduce the notion os swap loss OI which generalizes omniprediction in Section . In

---

[2]On a technical level, the omniprediction guarantee is made possible by post-processing the predictions after the loss is revealed. Importantly, the post-processing is a data-free univariate optimization that can be performed efficiently, based only on the loss function and not the data distribution.

Section B, we dicuss the relation and show separatiosn between various notions of omniprediction. All omitted proofs of formal claims are included in the Appendix.

## 2 Formal Setup: Swap Notions of Supervised Learning

We work in the *agnostic* learning setting, where we assume a data distribution $(\mathbf{x}, \mathbf{y}) \sim \mathcal{D}$ supported on $\mathcal{X} \times \{0, 1\}$. The objects of study in agnostic learning are real-valued hypotheses $h : \mathcal{X} \to \mathbb{R}$. Omniprediction and multicalibration study the special case of predictors $\tilde{p} : \mathcal{X} \to [0, 1]$ that map domain elements to probabilities. We denote the Bayes optimal predictor as $p^*(x) = \Pr[\mathbf{y} = 1|\mathbf{x} = x]$. As is standard in agnostic learning, we make no assumptions about the complexity of $p^* : \mathcal{X} \to [0, 1]$. While, in principle, predictors and hypotheses may take on continuous values in $\mathbb{R}$, we restrict our attention to functions supported on finitely-many values. We let $\mathrm{Im}(\tilde{p}) \subseteq [0, 1]$ denote the set of values taken on by $\tilde{p}(x)$.

Given a hypothesis $h : \mathcal{X} \to \mathbb{R}$, it will be useful to imagine drawing samples from $\mathcal{D}$ in two steps: first, we sample a prediction $\mathbf{v} \in \mathbb{R}$ according to the distribution of $h(\mathbf{x})$ under $\mathcal{D}$; then, we draw the random variables $(\mathbf{x}, \mathbf{y})$ according to the conditional distribution $\mathcal{D}|h(\mathbf{x}) = \mathbf{v}$. Accordingly, let $\mathcal{D}_h$ denote the distribution of $h(\mathbf{x})$ where $\mathbf{x} \sim \mathcal{D}$.

### 2.1 Swap Agnostic Learning

Swap agnostic learning is defined with respect to a loss function $\ell : \{0, 1\} \times \mathbb{R} \to \mathbb{R}$ and a hypothesis class $\mathcal{C} \subseteq \{c : \mathcal{X} \to \mathbb{R}\}$ and can be viewed as a game with two participants. The *predictor* plays first and selects a hypothesis $h : \mathcal{X} \to \mathbb{R}$. The *adversary* plays in response: for each level set $\{x \in \mathcal{X} : h(x) = v\}$, the adversary may choose a separate loss-minimizing hypothesis $c_v \in \mathcal{C}$.

**Definition 2.1** (Swap Agnostic Learning). *For a loss function $\ell$, hypothesis class $\mathcal{C}$, and error $\varepsilon \geq 0$, a hypothesis $h$ is a $(\ell, \mathcal{C}, \varepsilon)$-swap agnostic learner if*

$$\mathbf{E}[\ell(\mathbf{y}, h(\mathbf{x}))] \leq \underset{\mathbf{v} \sim \mathcal{D}_h}{\mathbf{E}} \left[ \min_{c_v \in \mathcal{C}} \ \mathbf{E}[\ell(\mathbf{y}, c_v(\mathbf{x}) \mid h(\mathbf{x}) = \mathbf{v}] \right] + \varepsilon. \tag{2}$$

We borrow nomenclature from online learning: a predictor using a swap agnostic learner $h$ has no incentive to "swap" any of their fixed predictions $h(\mathbf{x}) = v$ to predict according to $c_v \in \mathcal{C}$. Contrasting the requirement in (2) to that of agnostic learning in (1), we have switched the order of quantifiers, such that the minimization is taken after the expectation over the choice of $h(\mathbf{x}) = \mathbf{v}$. Swap agnostic learning strengthens standard agnostic learning, where the predictor only competes against the single best hypothesis $c \in \mathcal{C}$.

Indeed, swap agnostic learning seems to be a much more stringent condition. Provided the class $\mathcal{C}$ contains all constant functions, the adversary can simply imitate the predictor when $h(x) = v$ by choosing $c_v(x) = v$. For such $\mathcal{C}$, (2) implies that imitation is the best strategy for the adversary.

### 2.2 Swap Omniprediction

As in [15, 14], we observe that for any loss function $\ell$ and known distribution on outcomes $\mathbf{y} \sim \mathrm{Ber}(p)$ for $p \in [0, 1]$, there is an optimal action $k_\ell(p)$ that minimizes the expected loss $\ell$. Formally, we define the optimal post-processing of predictions by the function $k_\ell : [0, 1] \to \mathbb{R}$, which specifies the action that minimizes the expected loss.[3]

$$k_\ell(p) = \underset{t \in \mathbb{R}}{\arg\min} \ \underset{\mathbf{y} \sim \mathrm{Ber}(p)}{\mathbf{E}} [\ell(\mathbf{y}, t)]. \tag{3}$$

This observation is particularly powerful because the function $k_\ell$ is given by a simple, univariate optimization that can be used as a data-free post-processing procedure on top of a predictor $\tilde{p}$.

We recall the original notion of omniprediction proposed by [15], which requires a predictor to yield an agnostic learner simultaneously for every loss in a collection $\mathcal{L}$. Concretely, omniprediction requires that the function $h_\ell = k_\ell \circ \tilde{p}$ is an $(\ell, \mathcal{C}, \varepsilon)$-agnostic learner for every $\ell \in \mathcal{L}$.

---

[3]If there are multiple optima, we break ties arbitrarily.

**Definition 2.2** (Omnipredictor, [15]). *For a collection of loss functions $\mathcal{L}$, a hypothesis class $\mathcal{C}$, and error $\varepsilon > 0$, a predictor $\tilde{p} : \mathcal{X} \to [0,1]$ is an $(\mathcal{L}, \mathcal{C}, \varepsilon)$-omnipredictor if for every $\ell \in \mathcal{L}$,*

$$\mathbf{E}[\ell(\mathbf{y}, k_\ell(\tilde{p}(\mathbf{x})))] \leq \min_{c \in \mathcal{C}} \mathbf{E}[\ell(\mathbf{y}, c(\mathbf{x}))] + \varepsilon. \tag{4}$$

We propose a strengthened notion of Swap Omniprediction, where the adversary may choose both the loss $\ell_v \in \mathcal{L}$ and hypothesis $c_v \in \mathcal{C}$ based on the prediction $\tilde{p}(x) = v$.

**Definition 2.3** (Swap Omnipredictor). *For a collection of loss functions $\mathcal{L}$, a hypothesis class $\mathcal{C}$, and error $\varepsilon > 0$, a predictor $\tilde{p} : \mathcal{X} \to [0,1]$ is a $(\mathcal{L}, \mathcal{C}, \varepsilon)$-swap omnipredictor if for any assignment of loss functions $\{\ell_v \in \mathcal{L}\}_{v \in \mathrm{Im}(\tilde{p})}$,*

$$\mathop{\mathbf{E}}_{\mathbf{v} \sim \mathcal{D}_{\tilde{p}}} \Big[ \mathbf{E}[\ell_{\mathbf{v}}(\mathbf{y}, k_{\ell_{\mathbf{v}}}(v)) \mid \tilde{p}(\mathbf{x}) = \mathbf{v}] \Big] \leq \mathop{\mathbf{E}}_{\mathbf{v} \sim \mathcal{D}_{\tilde{p}}} \Big[ \min_{c_v \in \mathcal{C}} \mathbf{E}[\ell_{\mathbf{v}}(\mathbf{y}, c_v(\mathbf{x})) \mid \tilde{p}(\mathbf{x}) = \mathbf{v}] \Big] + \varepsilon. \tag{5}$$

Swap omniprediction gives the adversary considerable power. For instance, the special case where we restrict the adversary's choice of losses to be constant, $\ell_v = \ell$, realizes swap agnostic learning for $\ell$.

**Claim 2.4.** *If $\tilde{p}$ is a $(\mathcal{L}, \mathcal{C}, \varepsilon)$-swap omnipredictor, it is a $(\ell, \mathcal{C}, \varepsilon)$-swap agnostic learner for every $\ell \in \mathcal{L}$, and hence a $(\mathcal{L}, \mathcal{C}, \varepsilon)$-omnipredictor.*

Analogous to the standard notions, swap omniprediction implies swap agnostic learning for every $\ell$; however, swap omniprediction gives an even stronger guarantee since the adversary's loss $\ell_v$ may be chosen in response to the prediction $\tilde{p}(\mathbf{x}) = v$.

## 2.3 Swap Multicalibration

Multicalibration was introduced in the work of [20] as a notion of algorithmic fairness following [26], but has since seen broad application across many learning applications (e.g., [6, 15, 25, 18]). Informally, multicalibration requires predictions to appear calibrated, not simply overall, but also when we restrict our attention to subgroups within some broad collection $\mathcal{C}$. The formulation below appears in [14].

**Definition 2.5** (Multicalibration, [20]). *For a hypothesis class $\mathcal{C}$ and $\alpha \geq 0$, a predictor $\tilde{p} : \mathcal{X} \to [0,1]$ is $(\mathcal{C}, \alpha)$-multicalibrated if*

$$\max_{c \in \mathcal{C}} \mathop{\mathbf{E}}_{\mathbf{v} \sim \mathcal{D}_{\tilde{p}}} \Big[ \Big| \mathbf{E}[c(\mathbf{x})(\mathbf{y} - \mathbf{v}) \mid \tilde{p}(\mathbf{x}) = \mathbf{v}] \Big| \Big] \leq \alpha. \tag{6}$$

When $c : \mathcal{X} \to \{0,1\}$ is Boolean (and has sufficiently large measure), this definition says that conditioned on $c(\mathbf{x}) = 1$, the calibration violation is small, recovering the definition in [20].

Swap Multicalibration strengthens multicalibration, extending the pseudorandomness perspective on multicalibration developed in [6, 15]. Swap multicalibration requires that for the typical prediction $\mathbf{v} \sim \mathcal{D}_{\tilde{p}}$, no hypothesis in $c_v \in \mathcal{C}$ achieves good correlation with the residual labels $\mathbf{y} - \mathbf{v}$ over $\mathcal{D}$ conditioned on $\tilde{p}(\mathbf{x}) = \mathbf{v}$.

**Definition 2.6** (Swap Multicalibration). *For a hypothesis class $\mathcal{C}$ and $\alpha \geq 0$, a predictor $\tilde{p} : \mathcal{X} \to [0,1]$ is $(\mathcal{C}, \alpha)$-swap multicalibrated if*

$$\mathop{\mathbf{E}}_{\mathbf{v} \sim \mathcal{D}_{\tilde{p}}} \Big[ \max_{c_v \in \mathcal{C}} \Big| \mathbf{E}[c_v(\mathbf{x})(\mathbf{y} - \mathbf{v}) \mid \tilde{p}(\mathbf{x}) = \mathbf{v}] \Big| \Big] \leq \alpha. \tag{7}$$

Again, the difference between swap and standard multicalibration is in the order of quantifiers. The standard definition requires that for every $c \in \mathcal{C}$, the correlation $|\mathbf{E}_{\mathcal{D}_{|\mathbf{v}}}[c(\mathbf{x})(\mathbf{y} - \mathbf{v})]|$ achieved in small in expectation over $\mathbf{v} \sim \mathcal{D}_{\tilde{p}}$. Swap multicalibration considers the maximum of $|\mathbf{E}_{\mathcal{D}_{|\mathbf{v}}}[c(\mathbf{x})(\mathbf{y} - \mathbf{v})]|$ over all $c \in \mathcal{C}$ for each fixing of $\mathbf{v} = v$ and requires this to be small in expectation over $\mathbf{v} \sim \mathcal{D}_{\tilde{p}}$. It follows that swap multicalibration implies standard multicalibration.

**Claim 2.7.** *If $\tilde{p}$ is $(\mathcal{C}, \alpha)$-swap multicalibrated, it is $(\mathcal{C}, \alpha)$-multicalibrated.*

While swap multicalibration is nominally a stronger notion that than standard multicalibration, the complexity of achieving swap multicalibration is essentially the same as multicalibration. We show that starting from a $(\mathcal{C}, \alpha)$-multicalibrated predictor $\tilde{p}$, by suitably discretizing its values, we obtain a predictor that is close to $\tilde{p}$ and $(\mathcal{C}, \alpha')$-swap multicalibrated, with some degradation in the value of $\alpha'$.

**Definition 2.8.** *Let $\delta \in [0,1]$ so that $m = 1/\delta \in \mathbb{Z}$. Define $B_j = [(j-1)\delta, j\delta)$ for $j \in [m-1]$ and $B_m = [1-\delta, 1]$. Define the predictor $\bar{p}_\delta$ where for every $x$ such that $\tilde{p}(x) \in B_j$, $\bar{p}_\delta(x) = j\delta$.*

**Claim 2.9.** *Let $\tilde{p} : \mathcal{X} \to [0,1]$ be a $(\mathcal{C}, \alpha)$-multicalibrated. For any $\delta \in [0,1]$ where $1/\delta \in \mathbb{Z}$, the predictor $\bar{p}_\delta$ is $(\mathcal{C}, 2\sqrt{\alpha/\delta} + \delta)$-swap multicalibrated, and $\max_{x \in \mathcal{X}} |\tilde{p}(x) - \bar{p}_\delta(x)| \leq \delta$.*

While the bound of the theorem is valid for all $\delta$, the swap multicalibration guarantee is only meaningful when $\delta \geq \Omega(\alpha)$. Thus, by discretizing a $(\mathcal{C}, \alpha)$-multicalibrated predictor, we obtain a $(\mathcal{C}, O(\alpha^{1/3}))$-swap multicalibrated predictor. While this generic transformation suffers a polynomial loss in the accuracy parameter, such a loss may not be algorithmically necessary. As we argue in Lemma 3.8, known algorithms for achieving multicalibration [20, 15] actually guarantee swap multicalibration without any modification.

# 3 An equivalence: swap agnostic learning, omniprediction, multicalibration

Our main result is an equivalence between swap agnostic learning, swap omniprediction, and swap multicalibration. Concretely, this equivalence shows that swap agnostic learning for the squared error is sufficient to guarantee swap omniprediction for all (nice) convex loss functions. We begin with some preliminaries, then formally state and prove the equivalence. We conclude the section by showing how to use the existing framework for learning multicalibrated predictors to achieve swap agnostic learning for any convex loss.

**Nice loss functions.** For a loss function $\ell : \{0,1\} \times \mathbb{R} \to \mathbb{R}$, we extend $\ell$ linearly to allow the first argument to take values in the range $p \in [0,1]$ as:

$$\ell(p,t) = \mathop{\mathbb{E}}_{\mathbf{y} \sim \text{Ber}(p)} [\ell(\mathbf{y}, t)] = p \cdot \ell(1,t) + (1-p) \cdot \ell(0,t).$$

We say the loss function is *convex* if for $y \in \{0,1\}$, $\ell(y,t)$ is a convex function of $t$. By linearity, this convexity property holds for $\ell(p,t)$ for all $p$.

As in [14], for a loss $\ell$, we define the partial difference function $\partial\ell : \mathbb{R} \to \mathbb{R}$ as

$$\partial\ell(t) = \ell(1,t) - \ell(0,t).$$

We define a class of "nice" loss functions, which obey a minimal set of boundedness and Lipschitzness conditions.

**Definition 3.1.** *For a constant $B > 0$, a loss function is $B$-nice if there exists an interval $I_\ell \subseteq \mathbb{R}$ such that the following conditions hold:*

1. *(Optimality) If $\Pi_\ell : \mathbb{R} \to I_\ell$ denotes projection onto the interval $I_\ell$, then $\ell(p, \Pi_\ell(t)) \leq \ell(p,t)$ for all $t \in \mathbb{R}$ and $p \in [0,1]$.*

2. *(Lipschitzness) For $y \in \{0,1\}$ and $t \in I_\ell$, $\ell(y,t)$ is 1-Lipschitz as a function of $t$.*

3. *(Bounded difference) For $t \in I_\ell$, $|\partial\ell(t)| \leq B$.*

*The class of all $B$-nice loss functions by $\mathcal{L}(B)$. The subset of $B$-nice convex loss functions is denoted by $\mathcal{L}_{\text{cvx}}(B)$.*

Bounded loss functions generalize the idea of loss functions defined over a fixed interval of $\mathbb{R}$ (by optimality) and of bounded output range (by bounded difference). By optimality of nice loss functions, we may assume $k_\ell : [0,1] \to I_\ell$ since we do not increase the loss by projection onto $I_\ell$. Indeed, for convex losses $\ell$, the natural choice for $I_\ell$ is to take the interval $[k_\ell(0), k_\ell(1)]$. The bounded difference condition implies that $\ell$ is Lipschitz in its first argument, a property that will be useful.

**Lemma 3.2.** *For every $\ell \in \mathcal{L}(B)$ and $t_0 \in I_\ell$, the function $\ell(p, t_0)$ is $B$-Lipschitz as a function of $p$.*

**Concept classes.** For a concept class of functions $\mathcal{C} : \{c : \mathcal{X} \to \mathbb{R}\}$, we assume that $\mathcal{C}$ is closed under negation, and it contains the constant functions 0 and 1. Denoting $\|\mathcal{C}\|_\infty = \max |c(x)|$ over $c \in \mathcal{C}, x \in \mathcal{X}$, we say that $\mathcal{C}$ is bounded if $\|c\|_\infty \leq 1$. For $W \in \mathbb{R}^+$, let $\text{Lin}(\mathcal{C}, W)$ be all functions that can be expressed as a ($W$-sparse) linear combination of base concepts from $\mathcal{C}$,

$$c_w(x) = \sum_{c \in \mathcal{C}} w_c \cdot c(x), \quad \sum_{c \in \mathcal{C}} |w_c| \leq W.$$

Note that for bounded $\mathcal{C}$, the norm of linear combinations scales gracefully with the sparsity, $\|\mathrm{Lin}(\mathcal{C}, W)\|_\infty \leq W$. We define $\mathrm{Lin}(\mathcal{C})$ to be the set of all linear combinations with no restriction on the weights of the coefficients.

**Notation.** To simplify notation, we use the following shorthand for the data distribution $\mathcal{D}$ conditioned on $\tilde{p}(\mathbf{x}) = v$. For each $v \in \mathrm{Im}(\tilde{p})$, let $\mathcal{D}|_v$ denote the conditional distribution $\mathcal{D}|\tilde{p}(\mathbf{x}) = v$. Combined with earlier notation, the distribution $\mathcal{D}|_\mathbf{v}$ for $\mathbf{v} \sim \mathcal{D}_{\tilde{p}}$ is simply the data distribution $\mathcal{D}$. We also use the notation $(\tilde{p}(\mathbf{x}), \mathbf{y}) = (v, y)$ to indicate $\tilde{p}(\mathbf{x}) = v$ and $\mathbf{y} = y$.

### 3.1 Statement of Main Result

**Theorem 3.3.** *Let $\tilde{p}$ be a predictor, $\mathcal{C}$ be a bounded hypothesis class, and $\mathcal{L}_{\mathrm{cvx}}(B)$ be the class of $B$-nice convex loss functions. The following properties are equivalent:*[4]

1. *$\tilde{p}$ is $(\mathcal{C}, \alpha_1)$-swap multicalibrated.*

2. *$\tilde{p}$ is an $(\mathcal{L}_{\mathrm{cvx}}(B), \mathrm{Lin}(\mathcal{C}, W), O((W + B)\alpha_2))$-swap omnipredictor, for all $W \geq 1, B \geq 0$.*

3. *$\tilde{p}$ is an $(\ell_2, \mathrm{Lin}(\mathcal{C}, 2), \alpha_3)$-swap agnostic learner.*

In preparation for proving the theorem, we establish some preliminary results. We define a function $\alpha : \mathrm{Im}(\tilde{p}) \to [-1, 1]$ which measures the maximum correlation between $c \in \mathcal{C}$ and $\mathbf{y} - v$, conditioned on a prediction value $v \in \mathrm{Im}(\tilde{p})$. Let

$$\alpha(v) = \left| \max_{c_v \in \mathcal{C}} \mathop{\mathbf{E}}_{\mathcal{D}|_v} [c_v(\mathbf{x})(\mathbf{y} - v)] \right|.$$

Using this notation, $(\mathcal{C}, \alpha_0)$-swap multicalibration can be written as

$$\mathop{\mathbf{E}}_{\mathbf{v} \sim \mathcal{D}_{\tilde{p}}} [\alpha(\mathbf{v})] \leq \alpha_0.$$

We observe that swap multicalibration is closed under bounded linear combinations of $\mathcal{C}$, like with standard multicalibration.

**Claim 3.4.** *For every $h \in \mathrm{Lin}(\mathcal{C}, W)$ and $v \in \mathrm{Im}(\tilde{p})$, we have*

$$\max_{h \in \mathrm{Lin}(\mathcal{C}, W)} \left| \mathop{\mathbf{E}}_{\mathcal{D}} [h(\mathbf{x})(\mathbf{y} - v)|\tilde{p}(\mathbf{x}) = v] \right| \leq W \alpha(v).$$

*Let $p_v^* = \mathbf{E}[\mathbf{y}|\tilde{p}(\mathbf{x}) = v]$. Then*

$$|p_v^* - v| \leq \alpha(v). \tag{8}$$

Equation (8) follows since $1 \in \mathcal{C}$.

**Claim 3.5.** *For $h \in \mathrm{Lin}(\mathcal{C}, w)$, $v \in \mathrm{Im}(\tilde{p})$ and $y \in \{0, 1\}$, define the following conditional expectations:*

$$\mu(h : v) = \mathbf{E}[h(\mathbf{x})|\tilde{p}(\mathbf{x}) = v]$$
$$\mu(h : v, y) = \mathbf{E}[h(\mathbf{x})|(\tilde{p}(\mathbf{x}), \mathbf{y}) = (v, y)].$$

*Then for each $y \in \{0, 1\}$*

$$\Pr[\mathbf{y} = y|\tilde{p}(\mathbf{x}) = v] \, |\mu(h : v, y) - \mu(h : v)| \leq (W + 1)\alpha(v). \tag{9}$$

Next we show the following lemma, which shows that one can replace $h(\mathbf{x})$ by the constant $\Pi_\ell(\mu(h : v))$ without a large increase in the loss.

**Lemma 3.6.** *For all $h \in \mathrm{Lin}(\mathcal{C}, W)$, $v \in \mathrm{Im}(\tilde{p})$ and loss $\ell \in \mathcal{L}_{\mathrm{cvx}}(B)$, we have*

$$\mathop{\mathbf{E}}_{\mathcal{D}|_v} [\ell(\mathbf{y}, \Pi_\ell(\mu(h : v)))] \leq \mathop{\mathbf{E}}_{\mathcal{D}|_v} [\ell(\mathbf{y}, h(\mathbf{x}))] + 2(W + 1)\alpha(v). \tag{10}$$

---

[4]When we say these conditions are equivalent, we mean that they imply each other with parameters $\alpha_i$ that are polynomially related. The relations we derive result in at most a quadratic loss in parameters.

In the interest of space, we defer the proof of the Lemma to the Appendix.

Next we compare $\Pi_\ell(\mu(h : v))$ with $k_\ell(v)$. It is clear that the latter is better for minimizing loss when $\mathbf{y} \sim \mathrm{Ber}(v)$, by definition. We need to compare the losses when $\mathbf{y} \sim \mathrm{Ber}(p_v^*)$. But $p_v^*$ and $v$ are at most $\alpha(v)$ apart by Equation (8). Hence, by using Lipschitzness, one can infer that $k_\ell(v)$ is better than $\Pi_\ell(\mu(h : v))$ and hence $h(\mathbf{x})$. This is formalized in the following lemma and its proof.

**Lemma 3.7.** *For all $v \in \mathrm{Im}(\tilde{p})$, $\ell \in \mathcal{L}_{\mathsf{cvx}}(B)$ and $h \in \mathrm{Lin}(\mathcal{C}, W)$, we have*

$$\mathop{\mathbf{E}}_{\mathcal{D}|_v} \left[ \ell(\mathbf{y}, k_\ell(\tilde{p}(\mathbf{x}))) \right] \leq \mathop{\mathbf{E}}_{\mathcal{D}|_v} \left[ \ell(\mathbf{y}, h(\mathbf{x})) \right] + 2(W + B + 1)\alpha(v). \tag{11}$$

*Proof.* By the definition of $k_\ell$, $k_\ell(v)$ minimizes expected loss when $\mathbf{y} \sim \mathrm{Ber}(v)$, so

$$\ell(v, k_\ell(v)) \leq \ell(v, \Pi_\ell(\mu(h : v))) \tag{12}$$

On the other hand,

$$\mathop{\mathbf{E}}_{\mathcal{D}|_v} \left[ \ell(\mathbf{y}, t) \right] = \ell(p_v^*, t), \text{ where } p_v^* = \mathop{\mathbf{E}}_{\mathcal{D}|_v} \left[ \mathbf{y} \right].$$

Thus our goal is compare the losses $\ell(p_v^*, t)$ for $t = k_\ell(v)$ and $t = \Pi_\ell(\mu(h : v))$. Hence, applying Lemma 3.2 gives

$$\ell(p_v^*, k_\ell(v)) \leq \ell(v, k_\ell(v)) + \alpha(v)B$$
$$-\ell(p_v^*, \Pi_\ell(\mu(h : v)) \leq -\ell(v, \Pi_\ell(\mu(h : v)) + \alpha(v)B$$

Subtracting these inequalities and then using Equation (12) gives

$$\ell(p_v^*, k_\ell(v)) - \ell(p_v^*, \Pi_\ell(\mu(h : v)) \leq \ell(v, k_\ell(v)) - \ell(v, \Pi_\ell(\mu(h : v)) + 2B\alpha(v) \tag{13}$$
$$\leq 2\alpha(v)B. \tag{14}$$

We can now write

$$\mathop{\mathbf{E}}_{\mathcal{D}|_v} \left[ \ell(\mathbf{y}, k_\ell(v)) \right] = \ell(p_v^*, k_\ell(v))$$

$$\leq \ell(p_v^*, \Pi_\ell(\mu(h : v)) + 2\alpha(v)B \qquad \text{(by Equation (14))}$$
$$= \mathop{\mathbf{E}}_{\mathcal{D}|_v} \left[ \ell(\mathbf{y}, \Pi_\ell(\mu(h : v))) \right] + 2\alpha(v)B$$
$$\leq \mathop{\mathbf{E}}_{\mathcal{D}|_v} \left[ \ell(\mathbf{y}, h(\mathbf{x})) \right] + 2(W + 1)\alpha(v) + 2B\alpha(v). \qquad \text{(by Equation (10))}$$

∎

We now complete the proof of Theorem 3.3.

*Proof of Theorem 3.3.* (1) $\implies$ (2) Fix a $(\mathcal{C}, \alpha_1)$-swap multicalibrated predictor $\tilde{p}$. Fix a choice of loss functions $\{\ell_v \in \mathcal{L}_{\mathsf{cvx}}\}_{v \in \mathrm{Im}(f)}$ and hypotheses $\{h_v \in \mathcal{H}\}_{v \in \mathrm{Im}(f)}$. For each $v$, we apply Equation (11) with the loss $\ell = \ell_v$, hypothesis $h = h_v$ to get

$$\mathop{\mathbf{E}}_{\mathcal{D}|_v} \left[ \ell_v(\mathbf{y}, k_{\ell_v}(v)) \right] \leq \mathop{\mathbf{E}}_{\mathcal{D}|_v} \left[ \ell_v(\mathbf{y}, h_v(\mathbf{x})) \right] + 2(W + B + 1)\alpha(v).$$

We now take expectations over $\mathbf{v} \sim \mathcal{D}_{\tilde{p}}$, and use the $\mathbf{E}[\alpha(\mathbf{v})] \leq \alpha_1$ to derive the desired implication.

(2) $\implies$ (3) with $\alpha_3 = 7\alpha_2$ because $\ell_2$ is a $1/2$-nice loss function, so we plug in $B = 1/2$ and $W = 2$ into claim (2).

It remains to prove that (3) $\implies$ (1). We show the contrapositive, that if $\tilde{p}$ is not $\alpha_1$ multicalibrated, then $f = \tilde{p}$ is not a $(\ell_2, \mathrm{Lin}(\mathcal{C}, 2), \alpha_3)$-swap agnostic learner. By the definition of multicalibration, for every $v \in \mathrm{Im}(\tilde{p})$, there exist $c_v$ such that

$$\mathop{\mathbf{E}}_{\mathcal{D}|_v} \left[ c_v(\mathbf{x})(\mathbf{y} - \tilde{p}(\mathbf{x})) \right] = \alpha(v)$$

$$\mathop{\mathbf{E}}_{\mathcal{D}_{\tilde{p}}} [\alpha(\mathbf{v})] \geq \alpha_1.$$

By negating $c_v$ if needed, we may assume $\alpha(v) \geq 0$ for all $v$. We now define the updated hypothesis $h'$ where

$$h'(x) = v + \alpha(v)c_v(x) \text{ for } x \in \tilde{p}^{-1}(v)$$

---

**Algorithm 1** Swap Agnostic Learning via MCBoost

---

**Parameters:** loss $\ell$, hypothesis class $\mathcal{C}$, and $\varepsilon > 0$, let $\alpha = \mathrm{poly}(\varepsilon)$
**Given:** Dataset $S$ sampled from $\mathcal{D}$
**Run:**
$\tilde{p} \leftarrow \mathrm{MCBoost}_{\mathcal{C},\alpha}(S)$
$h_\ell \leftarrow k_\ell \circ \tilde{p}$
**Return:** $h_\ell$

---

A standard calculation (included in the Appendix) shows that
$$\mathop{\mathbf{E}}_{\mathcal{D}|_v}[(\mathbf{y} - v)^2] - \mathop{\mathbf{E}}_{\mathcal{D}|_v}[(\mathbf{y} - h'(\mathbf{x}))^2] \geq \alpha(v)^2.$$
Taking expectation over $\mathbf{v} \sim \mathcal{D}_{\tilde{p}}$, we have
$$\mathop{\mathbf{E}}_{\mathcal{D}_{\tilde{p}}}\left(\mathop{\mathbf{E}}_{\mathcal{D}|_\mathbf{v}}[(\mathbf{y} - v)^2] - \mathop{\mathbf{E}}_{\mathcal{D}|_\mathbf{v}}[(\mathbf{y} - h'(\mathbf{x}))^2]\right) \geq \mathop{\mathbf{E}}_{\mathcal{D}_{\tilde{p}}}[\alpha(\mathbf{v})^2] \geq \mathop{\mathbf{E}}_{\mathcal{D}_{\tilde{p}}}[\alpha(\mathbf{v})]^2 \geq \alpha_1^2.$$
It remains to show that $v + \alpha_v c_v(x) \in \mathrm{Lin}(\mathcal{C}, 2)$. Note that
$$\alpha_v = \mathop{\mathbf{E}}_{\mathcal{D}|_v}[c(\mathbf{x})(\mathbf{y} - v)v] \leq \max|c(\mathbf{x})|\max(|y - \tilde{p}(x)| \leq 1$$
since $c(\mathbf{x}), y - v \in [-1, 1]$. Hence $h'(x) = w_1 \cdot 1 + w_2 c(v)$ where $|w_1| + |w_2| \leq 2$. This contradicts the definition of an $(\ell_2, \mathrm{Lin}(\mathcal{C}, 2), \alpha_3)$-swap agnostic learner if $\alpha_3 < \alpha_1^2$. ∎

### 3.2 An algorithm for Swap Agnostic Learning

The equivalence from Theorem 3.3 suggests an immediate strategy for obtaining a swap agnostic learner. First, learn a swap multicalibrated predictor; then, return the predictor, post-processed to an optimal hypothesis according to $\ell$. While the MCBoost algorithm of [20] was designed to guarantee multicalibration, we observe that, in fact, it actually guarantees swap multicalibration.

**Lemma 3.8.** *Suppose the collection $\mathcal{C}$ has a weak agnostic learner, WAL. For any $\alpha > 0$, MCBoost makes at most $\mathrm{poly}(1/\alpha)$ calls to $\mathrm{WAL}_\alpha$, and returns a $(\mathcal{C}, \alpha)$-swap multicalibrated predictor.*

Weak agnostic learning is a basic supervised learning primitive used in boosting algorithms. Through the connection to boosting, weak agnostic learning is polynomial-time equivalent to agnostic learning, and inherits its data-efficiency (scaling with the VC-dimension of $\mathcal{C}$), but also its worst-case computational hardness [27]. Importantly, however, MCBoost reduces the problem of swap multicalibration (and thus, swap agnostic learning) to a standard agnostic learning task. We review MCBoost weak agnostic learning formally in the Appendix.

In all, we can combine the MCBoost algorithm for a class $\mathcal{C}$ with a specific loss $\ell$ to obtain a $(\ell, \mathcal{C})$-swap agnostic learner.

**Corollary 3.9** (Informal). *For any (nice) convex loss $\ell$, hypothesis class $\mathcal{C}$, and $\varepsilon > 0$, Algorithm 1 returns a $(\ell, \mathcal{C}, \varepsilon)$-swap agnostic learner from a sample of $m \leq \mathrm{VC}(\mathcal{C}) \cdot \mathrm{poly}(1/\varepsilon)$ data points drawn from $\mathcal{D}$, after making $\leq \mathrm{poly}(1/\varepsilon)$ calls to weak agnostic learner for $\mathcal{C}$.*

## 4 Beyond Swap Agnostic Learning: Swap Loss Outcome Indistinguishability

In this we introduce a unified notion of Swap Loss Outcome Indistinguishability, which captures all of the other notions of mutlicalibration and omniprediction defined so far. The notion builds on a line of work due to [6, 7], which propose the notion of *Outcome Indistinguishability* (OI) as a solution concept for supervised learning based on computational indistinguishability. In fact, the main result of [6] is an equivalence between OI and multicalibration. Despite the fact that OI is really multicalibration in disguise, the perspective has proved to be a useful technical perspective.

Key to this section is the prior work of [14]. This work proposes a new variant of OI, called *Loss OI*. The main result of [14] derives novel omniprediction guarantees from loss OI. Further, they show how to achieve loss OI using only calibration and multiaccuracy over a class of functions derived from the loss class $\mathcal{L}$ and hypothesis class $\mathcal{C}$. As we'll see, this class plays a role in the study of swap loss OI: swap loss OI is equivalent to multicalibration over the augmented class.

**Additional Preliminaries.** Intuitively, OI requires that outcomes sampled from the predictive model $\tilde{p}$ are indistinguishable from Nature's outcomes. Formally, we use $(\mathbf{x}, \mathbf{y}^*)$ to denote a sample from the true joint distribution over $\mathcal{X} \times \{0, 1\}$. Then, given a predictor $\tilde{p}$, we associate it with the random variable with $\mathbf{E}[\tilde{\mathbf{y}}|x] = \tilde{p}(x)$, i.e., where $\tilde{\mathbf{y}}|x \sim \text{Ber}(\tilde{p}(x))$. The variable $\tilde{\mathbf{y}}$ can be viewed as $\tilde{p}$'s simulation of Nature's label $\mathbf{y}^*$. In this section, we use $\mathcal{D}$ to denote the joint distribution $(\mathbf{x}, \mathbf{y}^*, \tilde{\mathbf{y}})$, where $\mathbf{E}[\mathbf{y}^*|x] = p^*(x)$ and $\mathbf{E}[\tilde{\mathbf{y}}|x] = \tilde{p}(x)$. While the joint distribution of $(\mathbf{y}^*, \tilde{\mathbf{y}})$ is not important to us, for simplicity we assume they are independent given $\mathbf{x} = x$.

## 4.1 Swap Loss OI

The notion of loss outcome indistinguishability was introduced in the recent work of [14] with the motivation of understanding omniprediction from the perspective of outcome indistinguishability [6]. Loss OI gives a strengthening of omniprediction. It requires predictors $\tilde{p}$ to fool a family $\mathcal{U}$ of statistical tests $u : \mathcal{X} \times [0, 1] \times \{0, 1\}$ that take a point $\mathbf{x} \in \mathcal{X}$, a prediction $\tilde{p}(\mathbf{x}) \in [0, 1]$ and a label $\mathbf{y} \in \{0, 1\}$ as their arguments. The goal is distinguish between the scenarios where $\mathbf{y} = \mathbf{y}^*$ is generated by *nature* versus where $\mathbf{y} = \tilde{\mathbf{y}}$ is a simulation of nature according to the predictor $\tilde{p}$. Formally, we require than for every $u \in \mathcal{U}$,

$$\mathbf{E}_{\mathcal{D}}[u(\mathbf{x}, \tilde{p}(\mathbf{x}), \mathbf{y}^*)] \approx_\varepsilon \mathbf{E}_{\mathcal{D}}[u(\mathbf{x}, \tilde{p}(\mathbf{x}), \tilde{\mathbf{y}})].$$

Loss OI specializes this to a specific family of tests arising in the analysis of omnipredictors.

**Definition 4.1** (Loss OI, [14]). *For a collection of loss functions $\mathcal{L}$, hypothesis class $\mathcal{C}$, and $\varepsilon \geq 0$, define the family of tests $\mathcal{U}(\mathcal{L}, \mathcal{C}) = \{u_{\ell,c}\}_{\ell \in \mathcal{L}, c \in \mathcal{C}}$ where*

$$u_{\ell,c}(x, v, y) = \ell(y, k_\ell(v)) - \ell(y, c(x)). \tag{15}$$

*A predictor $\tilde{p} : \mathcal{X} \to [0, 1]$ is $(\mathcal{L}, \mathcal{C}, \varepsilon)$-loss OI if for every $u \in \mathcal{U}(\mathcal{L}, \mathcal{C})$, it holds that*

$$\left| \mathbf{E}_{(\mathbf{x}, \mathbf{y}^*) \sim \mathcal{D}}[u(\mathbf{x}, \tilde{p}(\mathbf{x}), \mathbf{y}^*)] - \mathbf{E}_{(\mathbf{x}, \tilde{\mathbf{y}}) \sim \mathcal{D}(\tilde{p})}[u(\mathbf{x}, \tilde{p}(\mathbf{x}), \tilde{\mathbf{y}})] \right| \leq \varepsilon. \tag{16}$$

[14] show that loss-OI implies omniprediction.

**Lemma 4.2** (Proposition 4.5, [14]). *If the predictor $\tilde{p}$ is $(\mathcal{L}, \mathcal{C}, \varepsilon)$-loss OI, then it is an $(\mathcal{L}, \mathcal{C}, \varepsilon)$-omnipredictor.*

Indeed, if the expected value of $u$ is nonpositive for all $u \in \mathcal{U}(\mathcal{L}, \mathcal{C})$, then $\tilde{p}$ must achieve loss competitive with all $c \in \mathcal{C}$. The argument leverages the fact that $u$ must be nonpositive when $\tilde{\mathbf{y}} \sim \text{Ber}(\tilde{p}(\mathbf{x}))$—after all, in this world $\tilde{p}$ is the Bayes optimal. By indistinguishability, $\tilde{p}$ must also be optimal in the world where outcomes are drawn as $\mathbf{y}^*$. The converse, however, is not always true.

Next, we introduce swap loss OI, which allows the choice of distinguisher to depend on the predicted value.

**Definition 4.3** (Swap Loss OI). *For a collection of loss functions $\mathcal{L}$, hypothesis class $\mathcal{C}$ and $\varepsilon \geq 0$, for an assignment of loss functions $\{\ell_v \in \mathcal{L}\}_{v \in \text{Im}(\tilde{p})}$ and hypotheses $\{h_v \in \mathcal{H}\}_{v \in \text{Im}(\tilde{p})}$, denote $u_v = u_{\ell_v, c_v} \in \mathcal{U}(\mathcal{L}, \mathcal{C})$. A predictor $\tilde{p}$ is $(\mathcal{L}, \mathcal{C}, \alpha)$-swap loss OI if for all such assignments,*

$$\mathbf{E}_{\mathbf{v} \sim \mathcal{D}_{\tilde{p}}} \left| \mathbf{E}_{\mathcal{D}|_{\mathbf{v}}} [u_\mathbf{v}(\mathbf{x}, \mathbf{v}, \mathbf{y}^*) - u_\mathbf{v}(\mathbf{x}, \mathbf{v}, \tilde{\mathbf{y}})] \right| \leq \alpha.$$

The notion generalizes both swap omniprediction and loss-OI simultaneously.

**Lemma 4.4.** *If the predictor $\tilde{p}$ satisfies $(\mathcal{L}, \mathcal{C}, \alpha)$-swap loss OI, then*

- *it is an $(\mathcal{L}, \mathcal{C}, \alpha)$-swap omnipredictor.*

- *it is $(\mathcal{L}, \mathcal{C}, \alpha)$-loss OI.*

This Section continues as Section A in the Appendix, where we show the following characterization of loss OI.

**Theorem 4.5.** *Let $\mathcal{L}$ be a family of nice loss functions containing the squared loss. For any hypothesis class $\mathcal{C}$, a predictor satisfies $(\mathcal{L}, \mathcal{C})$-swap loss OI if and only if it is $(\partial \mathcal{L} \circ \mathcal{C})$-swap multicalibrated, where $\partial \mathcal{L} \circ \mathcal{C} = \{\partial \ell \circ c\}_{\ell \in \mathcal{L}, c \in \mathcal{C}}$.*

# 5 Related Work and Discussion

**Independent concurrent work.** The notion of strict multicalibration was defined in independent work of [8]. They connect this and other multigroup fairness definitions to various versions of the Szemeredi regularity lemma [28] and its weaker version due to Frieze and Kannan [11], following [29]. Their notions bears important similarities to swap multicalibration, but it is different. Like swap multicalibration, strict multicalibration involves switching the order of expectations and max. But they require a statistical closeness guarantee, whereas we only require a *first order guarantee*, that $c(\mathbf{x})$ be uncorrelated with $\mathbf{y} - \mathbf{v}$ conditioned on $\mathbf{v}$.

The independent work of [13] relates multicalibration to real-valued boosting ot minimize $\ell_2$ loss. The implication that $(\ell_2, \mathcal{C})$ swap-agnostic learning implies multicalibration follows from Part(1) of Theorem 3.2 in their paper. They prove that a violation of multicalibration leads to a better strategy for the adversary in the $(\ell_2, \mathcal{C})$-swap minimization game. They do not consider the notion of swap multicalibration, so their result is not a tight characterization unlike ours.

**Multi-group fairness and regret minimization.** Notions of multi-group fairness were introduced in the work of [20, 24] and [22], following [26]. The flagship notion of multicalibration has been extended to several other settings including multiclass predictions [7], real-valued labels [18, 21], online learning [18] and importance weights [17]. Alternate definitions and extensions of the standard notions of multicalibration have been proposed in [15, 16, 5, 8]. Multicalibration has also proved to have unexpected connections to many other domains, including computational indistinguishability [6], domain adaptation [25], and boosting [15, 13].

Our notions of swap loss minimization and swap omniprediction are inspired by notions of swap regret minimization in online learning [10, 9, 2]. The classic results of [10, 9] relate calibration and swap regret, whereas [2] show a generic way to convert a low external regret algorithm to ones with low internal and swap regret. The study of online learning and regret minimization is extensive, with deep connections to calibration and equilibria in game theory, we refer the readers to [4, 3] for comprehensive surveys. Recent work has established rich connections between online learning and regret minimization on one hand, and multigroup fairness notions on the other. Multicalibration has been considered in the online setting by [18], while [1, 12] relate multigroup fairness to questions in online learning.

**Conclusion.** Our work adds a new and significant connection in the theory of agnostic learning and multi-group fairness. This theoretical work relies on a few basic assumptions, including access to a representative unbiased data source and a weak agnostic learner for the collection $\mathcal{C}$. These assumptions, while standard in the supervised learning literature, should be interrogated before applying mutlicalibration as a notion of fairness in a practical setting.

Perhaps the most interesting aspect of our work is the connections it draws across different areas of learning theory. In particular, by borrowing the notion of swapping from online learning, we uncover surprisingly-powerful, but feasible solution concepts for supervised learning. The equivalence we draw between swap agnostic learning for the squared error and multicalibration with swap omniprediction *for all convex losses* highlights the power of simple goals like squared error minimization and calibration.

Our work gives a complete characterization of the notions we study at the "upper end" of strength (i.e., swap variants). A fascinating outstanding question to address in future research is whether there is a similar characterization of *standard* omniprediction in terms of multi-group fairness notions.

**Acknowledgements.** Omer Reingold is supported by the Simons Foundation Collaboration on the Theory of Algorithmic Fairness, the Simons Foundation investigator award 689988 and Sloan Foundation grant 2020-13941.

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

# A Equivalence of swap loss OI and swap multicalibration over augmented class

We show that $(\mathcal{L}, \mathcal{C})$-swap loss OI and $(\partial\mathcal{L} \circ \mathcal{C})$-swap multicalibration are equivalent for nice loss functions.

**Theorem A.1** (Formal statement of thm:swap-eq-inf). *Let $\mathcal{L} \subseteq \mathcal{L}(B)$ be a family of $B$-nice loss functions such that $\ell_2 \in \mathcal{L}$. Then $(\partial\mathcal{L} \circ \mathcal{C}, \alpha_1)$-swap multicalibration and $(\mathcal{L}, \mathcal{C}, \alpha_2)$-swap loss OI are equivalent.[5]*

In preparation for this, we first finish the proof of Lemma 4.4.

*Proof of Lemma 4.4.* The proof of Part (1) follows the proof of [14, Proposition 4.5], showing that loss OI implies omniprediction. By the definition of $k_{\ell_v}$, for every $x \in \mathcal{X}$ such that $\tilde{p}(x) = v$

$$
\mathop{\mathbf{E}}_{\tilde{\mathbf{y}} \sim \mathrm{Ber}(v)} u_v(x, v, \tilde{\mathbf{y}}) = \mathop{\mathbf{E}}_{\tilde{\mathbf{y}} \sim \mathrm{Ber}(v)} [\ell_v(\tilde{\mathbf{y}}, k_{\ell_v}(v)) - \ell_v(\tilde{\mathbf{y}}, c_v(x))]
$$
$$
= \ell_v(v, k_{\ell_v}(v)) - \ell_v(v, c_v(x))
$$
$$
\leq 0
$$

Hence this also holds in expectation under $\mathcal{D}|_v$, which only considers points where $\tilde{p}(\mathbf{x}) = v$:

$$
\mathop{\mathbf{E}}_{\mathcal{D}|_v} [u_v(\mathbf{x}, v, \tilde{\mathbf{y}})] \leq 0.
$$

Since $\tilde{p}$ satisfies swap loss OI, we deduce that

$$
\mathop{\mathbf{E}}_{\mathcal{D}|_v} [u_\mathbf{v}(\mathbf{x}, v, \mathbf{y}^*)] \leq \alpha(v)
$$

Taking expectations over $\mathbf{v} \sim \mathcal{D}_{\tilde{p}}$ and using the definition of $u_v$, we get

$$
\mathop{\mathbf{E}}_{\mathbf{v} \sim \mathcal{D}_{\tilde{p}}} \mathop{\mathbf{E}}_{\mathcal{D}|_\mathbf{v}} [\ell_\mathbf{v}(\mathbf{y}^*, k_{\ell_\mathbf{v}}(\mathbf{v})) - \ell_\mathbf{v}(\mathbf{y}^*, c_\mathbf{v}(\mathbf{x}))] = \mathop{\mathbf{E}}_{\mathbf{v} \sim \mathcal{D}_{\tilde{p}}} \mathop{\mathbf{E}}_{\mathcal{D}} [u_\mathbf{v}(\mathbf{x}, \mathbf{v}, \mathbf{y}^*)]
$$
$$
\leq \mathop{\mathbf{E}}_{\mathbf{v} \sim \mathcal{D}_{\tilde{p}}} [\alpha(\mathbf{v})] \leq \alpha
$$

Rearranging the outer inequality gives

$$
\mathop{\mathbf{E}}_{\mathbf{v} \sim \mathcal{D}_{\tilde{p}}} \mathop{\mathbf{E}}_{\mathcal{D}|_\mathbf{v}} [\ell_\mathbf{v}(\tilde{\mathbf{y}}, k_{\ell_\mathbf{v}}(\mathbf{v}))] \leq \mathop{\mathbf{E}}_{\mathbf{v} \sim \mathcal{D}_{\tilde{p}}} \mathop{\mathbf{E}}_{\mathcal{D}|_\mathbf{v}} [\ell_\mathbf{v}(\mathbf{y}^*, c_\mathbf{v}(\mathbf{x}))] + \alpha.
$$

Part (2) is implied by taking $\ell_v = \ell$ for every $v$. ∎

We use the following simple claim from [14].

**Claim A.2** (Lemma 4.8, [14]). *For random variables $\mathbf{y}_1, \mathbf{y}_2 \in \{0, 1\}$ and $t \in \mathbb{R}$,*

$$
\mathbf{E}[\ell(\mathbf{y}_1, t) - \ell(\mathbf{y}_2, t)] = \mathbf{E}[(\mathbf{y}_1 - \mathbf{y}_2)\partial\ell(t)]. \tag{17}
$$

We record two corollaries of this claim. These can respectively be seen as strengthenings of the two parts of Theorem [14, Theorem 4.9], which respectively characterized hypothesis OI in terms of multiaccuracy and decision OI in terms of calibration. We generalize these to the swap setting.

**Corollary A.3.** *For every choice of $\{\ell_v, c_v\}_{v \in \mathrm{Im}(\tilde{p})}$, we have*

$$
\mathop{\mathbf{E}}_{\mathbf{v} \sim \mathcal{D}_{\tilde{p}}} \left[ \left| \mathop{\mathbf{E}}_{\mathcal{D}|_\mathbf{v}} [\ell_\mathbf{v}(\mathbf{y}^*, c_\mathbf{v}(\mathbf{x})) - \ell_\mathbf{v}(\tilde{\mathbf{y}}, c_\mathbf{v}(\mathbf{x}))] \right| \right] = \mathop{\mathbf{E}}_{\mathbf{v} \sim \mathcal{D}_{\tilde{p}}} \left[ \left| \mathop{\mathbf{E}}_{\mathcal{D}|_\mathbf{v}} [(\mathbf{y}^* - \tilde{\mathbf{y}})\partial\ell_\mathbf{v} \circ c_\mathbf{v}(\mathbf{x})] \right| \right]. \tag{18}
$$

*Hence if $\tilde{p}$ is $(\partial\mathcal{L} \circ \mathcal{C}, \alpha)$-swap multicalibrated, then*

$$
\mathop{\mathbf{E}}_{\mathbf{v} \sim \mathcal{D}_{\tilde{p}}} \left[ \left| \mathop{\mathbf{E}}_{\mathcal{D}|_\mathbf{v}} [\ell_\mathbf{v}(\mathbf{y}^*, c_\mathbf{v}(\mathbf{x})) - \ell_\mathbf{v}(\tilde{\mathbf{y}}, c_\mathbf{v}(\mathbf{x}))] \right| \right] \leq \alpha.
$$

---

[5]Here equivalence means that there are reductions in either direction that lose a multiplicative factor of $(B + 1)$ in the error.

*Proof.* Equation (18) is derived by applying Equation (17) to the LHS. Assuming that $\tilde{p}$ is $(\partial\mathcal{L}\circ\mathcal{C}, \alpha)$-swap multicalibrated, we have

$$\mathop{\mathbf{E}}_{\mathbf{v}\sim\mathcal{D}_{\tilde{p}}}\left[\left|\mathop{\mathbf{E}}_{\mathcal{D}|_{\mathbf{v}}}\left[(\mathbf{y}^* - \tilde{\mathbf{y}})\partial\ell_{\mathbf{v}}\circ c_{\mathbf{v}}(\mathbf{x})\right]\right|\right] \le \mathop{\mathbf{E}}_{\mathbf{v}\sim\mathcal{D}_{\tilde{p}}}\left[\left|\max_{c'\in\partial\mathcal{L}\circ\mathcal{C}}\mathop{\mathbf{E}}_{\mathcal{D}|_{\mathbf{v}}}\left[(\mathbf{y}^* - \tilde{\mathbf{y}})c'(\mathbf{x})\right]\right|\right] \le \alpha.$$

∎

**Corollary A.4.** *Let $\{\ell_v\}_{v\in\mathrm{Im}(f)}$ be a collection of loss $B$-nice loss functions. Let $k(v) = k_{\ell_v}(v)$. If $\tilde{p}$ is $\alpha$-calibrated then*

$$\mathop{\mathbf{E}}_{\mathbf{v}\sim\mathcal{D}_{\tilde{p}}}\left[\left|\mathop{\mathbf{E}}_{\mathcal{D}|_{\mathbf{v}}}\left[\ell_{\mathbf{v}}(\mathbf{y}^*, k(\mathbf{v})) - \ell_{\mathbf{v}}(\tilde{\mathbf{y}}, k(\mathbf{v}))\right]\right|\right] \le B\alpha. \tag{19}$$

*Proof.* We have

$$\mathop{\mathbf{E}}_{\mathbf{v}\sim\mathcal{D}_{\tilde{p}}}\left[\left|\mathop{\mathbf{E}}_{\mathcal{D}|_{\mathbf{v}}}\left[\ell_{\mathbf{v}}(\mathbf{y}^*, k(\mathbf{v})) - \ell_{\mathbf{v}}(\tilde{\mathbf{y}}, k(\mathbf{v}))\right]\right|\right] = \mathop{\mathbf{E}}_{\mathbf{v}\sim\mathcal{D}_{\tilde{p}}}\left[\left|\mathop{\mathbf{E}}_{\mathcal{D}|_{\mathbf{v}}}\left[(\mathbf{y}^* - \mathbf{v})\partial\ell_{\mathbf{v}}(k(\mathbf{v}))\right]\right|\right]$$

$$= \mathop{\mathbf{E}}_{\mathbf{v}\sim\mathcal{D}_{\tilde{p}}}\left[|\partial\ell_{\mathbf{v}}(k(\mathbf{v}))|\left|\mathop{\mathbf{E}}_{\mathcal{D}|_{\mathbf{v}}}[\mathbf{y}^* - \mathbf{v}]\right|\right]$$

$$\le B\mathop{\mathbf{E}}_{\mathbf{v}\sim\mathcal{D}_{\tilde{p}}}\left[\left|\mathop{\mathbf{E}}_{\mathcal{D}|_{\mathbf{v}}}[\mathbf{y}^* - \mathbf{v}]\right|\right]$$

$$\le B\alpha.$$

where we use the fact that $k(v) \in I_\ell$, and so $|\partial\ell_v(k(v))| \le B$. ∎

Finally, we show the following key technical lemma which explains why the $\ell_2$ loss has a special role.

**Lemma A.5.** *If $\tilde{p}$ is $(\{\ell_2\}, \mathcal{C}, \alpha)$-swap OI, then it is $\alpha$-calibrated.*

*Proof.* Observe that $\ell_2(y, v) = (y - v)^2/2$ so $k_{\ell_2}(v) = v$. Hence,

$$u_{\ell_2,0}(x, v, y) = \ell_2(y, k_\ell(v)) - \ell_2(y, 0)$$
$$= ((y - v)^2 - y^2)/2$$
$$= -vy + v^2/2. \tag{20}$$

Recall that $\{0, 1\} \subset \mathcal{C}$. The implication of swap loss OI when we take $c_v = 0$ for all $v$ is that

$$\mathop{\mathbf{E}}_{\mathbf{v}\sim\mathcal{D}_{\tilde{p}}}\left[\left|\mathop{\mathbf{E}}_{\mathcal{D}|_{\mathbf{v}}}\left[u_{\ell_2,0}(\mathbf{x}, \mathbf{v}, \mathbf{y}^*) - u_{\ell_2,0}(\mathbf{x}, \mathbf{v}, \tilde{\mathbf{y}})\right]\right|\right] \le \alpha.$$

We can simplify the LHS using Equation (20) to derive

$$\mathop{\mathbf{E}}_{\mathbf{v}\sim\mathcal{D}_{\tilde{p}}}\left[\left|\mathop{\mathbf{E}}_{\mathcal{D}|_{\mathbf{v}}}\left[(-\mathbf{v}\mathbf{y}^* + \mathbf{v}^2/2) - (-\mathbf{v}\tilde{\mathbf{y}} + \mathbf{v}^2/2)\right]\right|\right] = \mathop{\mathbf{E}}_{\mathbf{v}\sim\mathcal{D}_{\tilde{p}}}\left[\left|\mathop{\mathbf{E}}_{\mathcal{D}|_{\mathbf{v}}}\left[\mathbf{v}(\mathbf{y}^* - \tilde{\mathbf{y}})\right]\right|\right]$$

$$= \mathop{\mathbf{E}}_{\mathbf{v}\sim\mathcal{D}_{\tilde{p}}}\left[\mathbf{v}\left|\mathop{\mathbf{E}}_{\mathcal{D}|_{\mathbf{v}}}[\tilde{\mathbf{y}} - \mathbf{y}^*]\right|\right] \le \alpha. \tag{21}$$

Considering the case where $c_v = 1$ for all $v$ gives

$$u_{\ell_2,1}(x, v, y) = \ell_2(y, k_\ell(v)) - \ell_2(y, 1)$$
$$= ((y - v)^2 - (1 - y)^2)/2$$
$$= (1 - v)y + (v^2 - 1)/2.$$

We derive the following implication of swap loss OI by taking $c_v = 0$ for all $v$:

$$\mathop{\mathbf{E}}_{\mathbf{v}\sim\mathcal{D}_{\tilde{p}}}\left[\left|\mathop{\mathbf{E}}_{\mathcal{D}|_{\mathbf{v}}}\left[u_{\ell_2,1}(\mathbf{x}, \mathbf{v}, \mathbf{y}^*) - u_{\ell_2,1}(\mathbf{x}, \mathbf{v}, \tilde{\mathbf{y}})\right]\right|\right] = \mathop{\mathbf{E}}_{\mathbf{v}\sim\mathcal{D}_{\tilde{p}}}\left[(1 - \mathbf{v})\left|\mathop{\mathbf{E}}_{\mathcal{D}|_{\mathbf{v}}}[\tilde{\mathbf{y}} - \mathbf{y}^*]\right|\right] \le \alpha \tag{22}$$

Adding the bounds from Equations (21) and (22) we get

$$\mathop{\mathbf{E}}_{\mathbf{v}\sim\mathcal{D}_{\tilde{p}}}\left[\left|\mathop{\mathbf{E}}_{\mathcal{D}|_{\mathbf{v}}}[\mathbf{v} - \mathbf{y}^*]\right|\right] = \mathop{\mathbf{E}}_{\mathbf{v}\sim\mathcal{D}_{\tilde{p}}}\left[\left|\mathop{\mathbf{E}}_{\mathcal{D}|_{\mathbf{v}}}[\tilde{\mathbf{y}} - \mathbf{y}^*]\right|\right] \le \alpha$$

∎

We can now complete the proof of Theorem A.1.

*Proof of Theorem A.1.* We first show the forward implication, that swap multicalibration implies swap loss OI.

Since $\ell_2 \in \mathcal{L}$ and $1 \in \mathcal{C}$, we have $\partial \ell_2 \circ 1 = 1 \in \partial \mathcal{L} \circ \mathcal{C}$. This implies that $\tilde{p}$ is $\alpha$-mulitcalibrated, since

$$\mathop{\mathbf{E}}_{\mathbf{v} \sim \mathcal{D}_{\tilde{p}}} \left[ \left| \mathop{\mathbf{E}}_{\mathcal{D}|_{\mathbf{v}}} [1(\mathbf{y} - \mathbf{v})] \right| \right] \leq \mathop{\mathbf{E}}_{\mathbf{v} \sim \mathcal{D}_{\tilde{p}}} \left[ \max_{c \in \mathcal{C}} \left| \mathop{\mathbf{E}}_{\mathcal{D}|_{\mathbf{v}}} [c(\mathbf{x})(\mathbf{y} - \mathbf{v})] \right| \right] \leq \alpha.$$

Consider any collection of losses $\{\ell_v\}_{v \in \text{Im}(\tilde{p})}$. Applying Corollary A.4, we have

$$\mathop{\mathbf{E}}_{\mathbf{v} \sim \mathcal{D}_{\tilde{p}}} \left[ \left| \mathop{\mathbf{E}}_{\mathcal{D}|_{\mathbf{v}}} [\ell_{\mathbf{v}}(\mathbf{y}^*, k(\mathbf{v})) - \ell_{\mathbf{v}}(\tilde{\mathbf{y}}, k(\mathbf{v}))] \right| \right] \leq B\alpha.$$

On the other hand, by Corollary A.3, we have for every choice of $\{\ell_v, c_v\}_{v \in \text{Im}(\tilde{p})}$,

$$\mathop{\mathbf{E}}_{\mathbf{v} \sim \mathcal{D}_{\tilde{p}}} \left[ \left| \mathop{\mathbf{E}}_{\mathcal{D}|_{\mathbf{v}}} [\ell_{\mathbf{v}}(\mathbf{y}^*, c_{\mathbf{v}}(\mathbf{x})) - \ell_{\mathbf{v}}(\tilde{\mathbf{y}}, c_{\mathbf{v}}(\mathbf{x}))] \right| \right] \leq \alpha.$$

Hence for any choice of $\{u_v\}_{v \in \text{Im}(\tilde{p})}$ we can bound

$$\mathop{\mathbf{E}}_{\mathbf{v} \sim \mathcal{D}_{\tilde{p}}} \left| \mathop{\mathbf{E}}_{\mathcal{D}|_{\mathbf{v}}} [u_{\mathbf{v}}(\mathbf{x}, \mathbf{v}, \mathbf{y}^*) - u_{\mathbf{v}}(\mathbf{x}, \mathbf{v}, \tilde{\mathbf{y}})] \right|$$

$$\leq \mathop{\mathbf{E}}_{\mathbf{v} \sim \mathcal{D}_{\tilde{p}}} \left[ \left| \mathop{\mathbf{E}}_{\mathcal{D}|_{\mathbf{v}}} [\ell_{\mathbf{v}}(\mathbf{y}^*, k(\mathbf{v})) - \ell_{\mathbf{v}}(\tilde{\mathbf{y}}, k(\mathbf{v}))] \right| + \left| \mathop{\mathbf{E}}_{\mathcal{D}|_{\mathbf{v}}} [\ell_{\mathbf{v}}(\mathbf{y}^*, c_{\mathbf{v}}(\mathbf{x})) - \ell_{\mathbf{v}}(\tilde{\mathbf{y}}, c_{\mathbf{v}}(\mathbf{x}))] \right| \right]$$

$$\leq (B+1)\alpha$$

which shows that $\tilde{p}$ satisfies swap loss OI with $\alpha_2 = (D+1)\alpha_1$.

Next we show the reverse implication: if $\tilde{p}$ satisfies $(\mathcal{L}, \mathcal{C}, \alpha_2)$-swap loss OI, then it satisfies $(\partial \mathcal{L} \circ \mathcal{C}, \alpha_1)$-swap multicalibration. The first step is to observe that by lemma A.5, since $\ell_2 \in \mathcal{L}$, the predictor $\tilde{p}$ is $\alpha_2$ calibrated. Since any $\ell \in \mathcal{L}$ is $B$-nice, we have

$$\mathop{\mathbf{E}}_{\mathbf{v} \sim \mathcal{D}_{\tilde{p}}} \left[ \left| \mathop{\mathbf{E}}_{\mathcal{D}|_{\mathbf{v}}} [\ell_{\mathbf{v}}(\mathbf{y}^*, k(\mathbf{v})) - \ell_{\mathbf{v}}(\tilde{\mathbf{y}}, k(\mathbf{v}))] \right| \right] = \mathop{\mathbf{E}}_{\mathbf{v} \sim \mathcal{D}_{\tilde{p}}} \left[ \left| \mathop{\mathbf{E}}_{\mathcal{D}|_{\mathbf{v}}} [(\mathbf{y}^* - \tilde{\mathbf{y}})k(\mathbf{v})] \right| \right] \leq B\alpha_2.$$

For any $\{\ell_v, c_v\}_{v \in \text{Im}(f)}$, since

$$u_v(x, v, y) = \ell_v(y, k_\ell(v)) + \ell_v(y, c_v(x))$$

we can write

$$\mathop{\mathbf{E}}_{\mathbf{v} \sim \mathcal{D}_{\tilde{p}}} \left[ \left| \mathop{\mathbf{E}}_{\mathcal{D}|_{\mathbf{v}}} [\ell_{\mathbf{v}}(\mathbf{y}^*, c(\mathbf{x})) - \ell_{\mathbf{v}}(\tilde{\mathbf{y}}, c(\mathbf{x}))] \right| \right]$$

$$\leq \mathop{\mathbf{E}}_{\mathbf{v} \sim \mathcal{D}_{\tilde{p}}} \left[ \left| \mathop{\mathbf{E}}_{\mathcal{D}|_{\mathbf{v}}} [u_{\mathbf{v}}(\mathbf{x}, \mathbf{v}, \mathbf{y}^*) - u_{\mathbf{v}}(\mathbf{x}, \mathbf{v}, \tilde{\mathbf{y}})] \right| + \left| \mathop{\mathbf{E}}_{\mathcal{D}|_{\mathbf{v}}} [\ell_{\mathbf{v}}(\mathbf{y}^*, k(\mathbf{v})) - \ell_{\mathbf{v}}(\tilde{\mathbf{y}}, k(\mathbf{v}))] \right| \right]$$

$$\leq (B+1)\alpha_2.$$

But by Equation (18), the LHS can be written as

$$\mathop{\mathbf{E}}_{\mathbf{v} \sim \mathcal{D}_{\tilde{p}}} \left[ \left| \mathop{\mathbf{E}}_{\mathcal{D}|_{\mathbf{v}}} [\ell_{\mathbf{v}}(\mathbf{y}^*, c(\mathbf{x})) - \ell_{\mathbf{v}}(\tilde{\mathbf{y}}, c(\mathbf{x}))] \right| \right] = \mathop{\mathbf{E}}_{\mathbf{v} \sim \mathcal{D}_{\tilde{p}}} \left[ \left| \mathop{\mathbf{E}}_{\mathcal{D}|_{\mathbf{v}}} [\partial \ell_{\mathbf{v}} \circ c_{\mathbf{v}}(\mathbf{x})(\mathbf{y}^* - \mathbf{v})] \right| \right]$$

This shows that $\tilde{p}$ is $(\partial \mathcal{L} \circ \mathcal{C}, (B+1)\alpha_2)$-swap multicalibrated. ∎

# B  Relating notions of omniprediction

In this work, we have discussed the four different notions of omniprediction defined to date.

    00)  Omniprediction, as originally defined by [15].

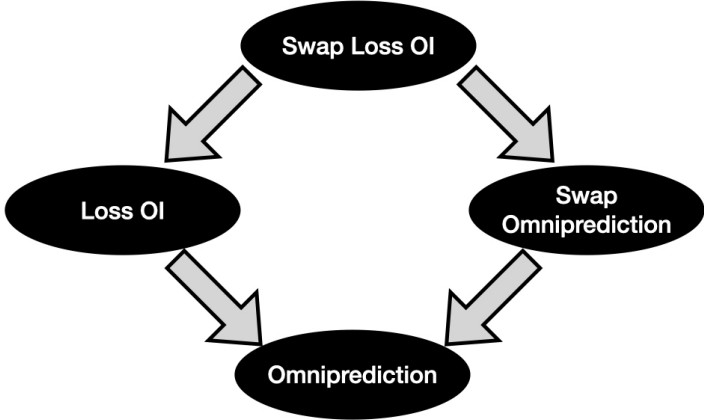

Figure 1: Relation between notions of omniprediction

01) Loss OI, from [14].

10) Swap omniprediction.

11) Swap Loss OI.

In order to compare them, we can ask which of these notions implies the other for any fixed choice of loss class $\mathcal{L}$ and hypothesis class $\mathcal{C}$.

- Loss OI implies omniprediction by [14, Proposition 4.5].
- Swap omniprediction implies omniprediction by Claim 2.4.
- Swap loss OI implies both loss OI and swap multicalibration by Lemma 4.4.

These relationships are summarized in Figure 1.

Further, this picture captures all the implications that hold for all $(\mathcal{L}, \mathcal{C})$. Next, we show that for any implication not drawn in the diagram, there exists some (natural) choice of $(\mathcal{L}, \mathcal{C})$, where the implication does not hold. In particular, we prove that neither loss OI nor swap omniprediction implies the other for all $(\mathcal{L}, \mathcal{C})$. This separates these notions from swap loss OI, since swap loss OI implies both these notions.[6] By similar reasoning, it separates omniprediction from both these loss OI and swap omnipredicition, since omniprediction is implied by either of them.

**Swap omniprediction does not imply loss OI.** We prove this non-implication using a counterexample used in [14]. In particular, they show that omniprediction does not imply loss OI [14, Theorem 4.6], and the same example in fact shows that swap omniprediction does not imply loss OI. In their example, we have $\mathcal{D}$ on $\{\pm 1\}^3 \times [0, 1]$ where the marginal on $\{\pm 1\}^3$ is uniform, and $p^*(x) = (1 + x_1 x_2 x_3)/2$, whereas $\tilde{p}(x) = 1/2$ for all $x$. We take $\mathcal{C} = \{1, x_1, x_2, x_3\}$. Since $\tilde{p} = 1/2$ is constant, it is easy to check that $\tilde{p} - p^* = -x_1 x_2 x_3/2$ is uncorrelated with $\mathcal{C}$. Hence $\tilde{p}$ satisfies swap multicalibration (which is the same as multicalibration or even multiaccuracy in this setting where $\tilde{p}$ is constant). Hence by Theorem 3.3, $\tilde{p}$ is an $(\mathcal{L}_{\mathsf{cvx}}(1), \mathrm{Lin}_{\mathcal{C}}, 0)$-swap omnipredictor. [14, Theorem 4.6] prove that $\tilde{p}$ is not loss OI for the $\ell_4$ loss. Hence we have the following result.

**Lemma B.1.** *The predictor $\tilde{p}$ is $(\mathcal{C}, 0)$-swap multicalibrated and hence it is a $(\{\ell_4\}, \mathrm{Lin}(\mathcal{C}), 0)$-swap omnipredictor. But it is not $(\{\ell_4\}, \mathrm{Lin}(\mathcal{C}, 1), \varepsilon)$-loss OI for $\varepsilon < 4/9$.*

We remark that the construction extends to all $\ell_p$ losses for even $p > 2$. Hence even for convex losses, the notions of swap omniprediction are loss-OI seem incomparable.

---

[6]For instance if loss OI implied swap loss OI, it would also imply swap omniprediction, which our claim shows it does not.

| $x = (x_1, x_2)$ | $p^*(x)$ | $\tilde{p}(x)$ |
|:---:|:---:|:---:|
| $(-1, -1)$ | $0$ | $\frac{1}{8}$ |
| $(+1, -1)$ | $\frac{1}{4}$ | $\frac{1}{8}$ |
| $(-1, +1)$ | $1$ | $\frac{7}{8}$ |
| $(+1, +1)$ | $\frac{3}{4}$ | $\frac{7}{8}$ |

Table 1: Separating loss-OI and swap-resilient omniprediction

**Loss OI does not imply swap omniprediction.**  Next we construct an example showing that loss OI need not imply swap omniprediction. We consider the set of all GLM losses defined below, which contain common losses including the squared loss and the logistic loss.

**Definition B.2.** *Let $g : \mathbb{R} \to \mathbb{R}$ be a convex, differentiable function such that $[0, 1] \subseteq \mathrm{Im}(g')$. Define its matching loss to be $\ell_g = g(t) - yt$. Define $\mathcal{L}_{\mathsf{GLM}} = \{\ell_g\}$ be the set of all such loss functions.*

[14] shows a general decomposition result that reduces achieving loss OI to a calibration condition and a multiaccuracy condition. Whereas arbitrary losses might require multiaccuracy for the more powerful class $\partial \mathcal{L} \circ \mathcal{C}$, for $\mathcal{L}_{\mathsf{GLM}}$, $\partial \mathcal{L}_{\mathsf{GLM}} \circ \mathcal{C} = \mathcal{C}$. This is formalized in the following result.

**Lemma B.3** (Theorem 5.3, [14]). *If $\tilde{p}$ is $\varepsilon_1$-calibrated and $(\mathcal{C}, \varepsilon_2)$-multiaccurate, then it is $(\mathcal{L}_{\mathsf{GLM}}, \mathrm{Lin}(\mathcal{C}, W), \delta)$-loss OI for $\delta = \varepsilon_1 + W\varepsilon_2$.*

In light of the above result, it suffices to find a predictor that is calibrated and multiaccurate (and hence satisfies loss OI), but not multicalibrated, hence not swap multicalibrated. By Theorem 3.3 it is not an $(\{\ell_2\}, \mathrm{Lin}_{\mathcal{C}}, \delta)$-swap omnipredictor for $\delta$ less than some constant.

Let us define the predictors $p^*, \tilde{p} : \{\pm 1\}^2 \to [0, 1]$ as below. We use these to show a separation between loss OI and swap omniprediction.

**Lemma B.4.** *Consider the distribution $\mathcal{D}$ on $\{\pm 1\}^2 \times \{0, 1\}$ where the marginal on $\{\pm 1\}^2$ is uniform and $\mathbf{E}[\mathbf{y}|x] = p^*(x)$. Let $\mathcal{C} = \{1, x_1, x_2\}$.*

1. *$\tilde{p} \in \mathrm{Lin}(\mathcal{C}, 1)$. Moreover, it minimizes the squared error over all hypotheses from $\mathrm{Lin}(\mathcal{C})$.*

2. *$\tilde{p}$ is perfectly calibrated and $(\mathcal{C}, 0)$-multiaccurate. So it is $(\mathcal{L}_{\mathsf{GLM}}, \mathrm{Lin}(\mathcal{C}), 0)$-loss OI.*

3. *$\tilde{p}$ is not $(\mathcal{C}, \alpha)$-multicalibrated for $\alpha < 1/8$. It is not $(\ell_2, \mathrm{Lin}(\mathcal{C}), \delta)$-swap agnostic learner for $\delta < 1/64$.*

*Proof.* We compute Fourier expansions for the two predictors:

$$p^*(x) = \frac{1}{8}(4 + 3x_2 - x_1 x_2) \tag{23}$$

$$\tilde{p}(x) = \frac{1}{8}(4 + 3x_2) \tag{24}$$

This shows that $\tilde{p} \in \mathrm{Lin}(\mathcal{C})$, and moreover that it is the optimal approximation to $p^*$ in $\mathrm{Lin}(\mathcal{C})$, as it is the projection of $p^*$ onto $\mathrm{Lin}(\mathcal{C})$. This shows that $\tilde{p}$ is an $(\ell_2, \mathrm{Lin}(\mathcal{C}), 0)$-agnostic learner.

It is easy to check that $\tilde{p}$ is perfectly calibrated. It is $(\mathcal{C}, 0)$-multiaccurate, since it is the projection of $p^*$ onto $\mathrm{Lin}(\mathcal{C})$, so $\tilde{p} - p^*$ is orthogonal to $\mathrm{Lin}(\mathcal{C})$. Hence we can apply Lemma B.3 to conclude that it is $(\mathcal{L}_{\mathsf{GLM}}, \mathrm{Lin}(C), 0)$-loss OI, where $\mathcal{L}_{\mathsf{GLM}}$ which contains the squared loss.

To show that $\tilde{p}$ is not swap-agnostic, we observe that conditioning on the value of $\tilde{p}(\mathbf{x}) = (4 + 3x_2)/8$ is equivalent to conditioning on $x_2 \in \{\pm 1\}$. For each value of $x_2$, the restriction of $p^*$ which is now linear in $x_1$ belongs to $\mathrm{Lin}(\mathcal{C})$. Indeed if we condition on $\tilde{p}(x) = 1/8$ so that $x_2 = -1$, we have

$$p^*(x) = \frac{1}{2} - \frac{3}{8} + \frac{1}{8}x_1 = \frac{1 + x_1}{8}.$$

Conditioned on $\tilde{p}(x) = 7/8$ so that $x_2 = 1$, we have

$$p^*(x) = \frac{1}{2} + \frac{3}{8} - \frac{1}{8}x_1 = \frac{7 - x_1}{8}.$$

Hence we have

$$\mathop{\mathbf{E}}_{v\sim\mathcal{D}_{\tilde{p}}}\left[\left\|\min_{h\in\mathrm{Lin}(\mathcal{C})}\mathbf{E}[(\mathbf{y}-h(\mathbf{x}))^2|f(\mathbf{x})=v]\right\|\right]=\mathbf{E}[(y-p^*(x))^2]=\mathrm{Var}[\mathbf{y}],$$

whereas the variance decomposition of squared loss gives

$$\mathbf{E}[(\mathbf{y}-\tilde{p}(\mathbf{x}))^2]=\mathbf{E}[(\mathbf{y}-p^*(\mathbf{x}))^2]+\mathbf{E}[(p^*(\mathbf{x})-\tilde{p}(\mathbf{x}))^2]$$
$$=\mathrm{Var}[\mathbf{y}]+\frac{1}{64}\,\mathbf{E}[(x_1x_2)^2]$$
$$=\mathrm{Var}[\mathbf{y}]+\frac{1}{64}.$$

Hence $\tilde{p}$ is not a $(\ell_2,\mathrm{Lin}(\mathcal{C}),\delta)$-swap agnostic learner for $\delta<1/64$.

To see that $f$ is not multicalibrated for small $\alpha$, observe that conditioned on $x_2\in\{\pm1\}$, the correlation between $x_1$ and $\tilde{p}-p^*$ is $1/8$. ∎

Note that item (1) above separates swap omniprediction from omniprediction and agnostic learning. This separation can also be derived from [15, Theorem 7.5] which separated (standard) omniprediction from agnostic learning, since swap omniprediction implies standard omniprediction.

**Comparing notions for GLM losses.** When we restrict our attention to $\mathcal{L}_{\mathsf{GLM}}$, in fact, the notions of swap loss OI and swap omniprediction are equivalent. The key observation here is that $\partial\mathcal{L}_{\mathsf{GLM}}\circ\mathcal{C}=\mathcal{C}$, as shown in [14]. Paired with Theorem 3.3 and Theorem A.1 (proved next), we obtain the following collapse.

**Claim B.5.** *The notions of $(\mathcal{L}_{\mathsf{GLM}},\mathcal{C},\alpha_1)$-swap loss OI and $(\mathcal{L}_{\mathsf{GLM}},\mathcal{C},\alpha_2)$-swap omniprediction are equivalent.*

*Proof.* To see this, note that by Theorem A.1, $(\mathcal{L}_{\mathsf{GLM}},\mathcal{C},\alpha_1)$-swap loss OI is equivalent to $(\partial\mathcal{L}_{\mathsf{GLM}}\circ\mathcal{C},\alpha_1')$-swap multicalibration. We know from Theorem 3.3 that this is also equivalent to $(\partial\mathcal{L}_{\mathsf{GLM}}\circ\mathcal{C},\alpha_2)$-swap omniprediction. So, by the fact that $\partial\mathcal{L}_{\mathsf{GLM}}\circ\mathcal{C}=\mathcal{C}$, we have the claimed equivalence. ∎

Finally, we know that loss OI implies omniprediction for $\mathcal{L}_{\mathsf{GLM}}$, since this holds true for all $\mathcal{L}$. We do not know if these notions are equivalent for $\mathcal{L}_{\mathsf{GLM}}$, since the construction in Lemma B.1 used the $\ell_4$ loss which does not belong to $\mathcal{L}_{\mathsf{GLM}}$.

## C  Omitted Proofs

We define the multicalibration error of $\tilde{p}$ wrt $\mathcal{C}$ under $\mathcal{D}$ as

$$\mathsf{MCE}_{\mathcal{D}}(f,\mathcal{C})=\max_{c\in\mathcal{C}}\mathop{\mathbf{E}}_{\mathbf{v}\sim\mathcal{D}_{\tilde{p}}}\left[\left\|\mathop{\mathbf{E}}_{\mathcal{D}|\mathbf{v}}[c(\mathbf{x})(\mathbf{y}-\mathbf{v})]\right\|\right].$$

We define the swap multicalibration error of $\tilde{p}$ wrt $\mathcal{C}$ under $\mathcal{D}$ as

$$\mathsf{sMCE}_{\mathcal{D}}(\tilde{p},\mathcal{C})=\mathop{\mathbf{E}}_{\mathbf{v}\sim\mathcal{D}_{\tilde{p}}}\left[\max_{c\in\mathcal{C}}\left|\mathop{\mathbf{E}}_{\mathcal{D}|\mathbf{v}}[c(\mathbf{x})(\mathbf{y}-\mathbf{v})]\right|\right]$$

### C.1  Properties of Swap Notions of Supervised Learning

*Proof of Claim 2.4.* We let $\ell_v=\ell$ for all $v\in\mathrm{Im}(\tilde{p})$, so that $k(v)=k_\ell(v)$. We pick the hypothesis

$$h_v=\arg\min_{h\in\mathcal{H}}\mathop{\mathbf{E}}_{\mathcal{D}|v}[\ell(\mathbf{y},h(\mathbf{x}))]$$

The swap omniprediction guarantee reduces to

$$\mathop{\mathbf{E}}_{\mathbf{v}\in\mathcal{D}_{\tilde{p}}}[\mathop{\mathbf{E}}_{\mathcal{D}|\mathbf{v}}[\ell(\mathbf{y},k_\ell(\mathbf{v}))]=\mathop{\mathbf{E}}_{\mathcal{D}}[\ell(\mathbf{y},k_\ell(\tilde{p}(\mathbf{x})))]\le\mathop{\mathbf{E}}_{\mathbf{v}\sim\mathcal{D}_{\tilde{p}}}\min_{h\in\mathcal{H}}\mathop{\mathbf{E}}_{\mathcal{D}|\mathbf{v}}[\ell(\mathbf{y},h(\mathbf{x}))]+\delta.$$

This implies that $f=k_\ell\circ\tilde{p}$ is a swap agnostic learner for every $\ell\in\mathcal{L}$ since we allow the choice of $h$ to depend on $\tilde{p}(\mathbf{x})$ which is more informative than $f(\mathbf{x})=k_\ell(\tilde{p}(\mathbf{x}))$. ∎

*Proof of Claim 2.7.* We have

$$\mathsf{sMCE}_{\mathcal{D}}(\tilde{p}, \mathcal{C}) = \mathop{\mathbf{E}}_{\mathbf{v} \sim \mathcal{D}_{\tilde{p}}} \left[ \max_{c \in \mathcal{C}} \left| \mathop{\mathbf{E}}_{\mathcal{D}|_{\mathbf{v}}} [c(\mathbf{x})(\mathbf{y}^* - \mathbf{v})] \right| \right]$$

$$\geq \max_{c \in \mathcal{C}} \mathop{\mathbf{E}}_{\mathbf{v} \sim \mathcal{D}_{\tilde{p}}} \left[ \left| \mathop{\mathbf{E}}_{\mathcal{D}|_{\mathbf{v}}} [c(\mathbf{x})(\mathbf{y} - \mathbf{v})] \right| \right] = \mathsf{MCE}_{\mathcal{D}}(\tilde{p}, \mathcal{C})$$

since the expectation of the max is higher than the max of expectations. Bounding the RHS by $\alpha$ is equivalent to $(\mathcal{C}, \alpha)$-multicalibration. ∎

*Proof of Claim 2.9.* The $\ell_\infty$ bound is immediate from the definition of $\bar{p}$. We bound the swap multicalibration error of $tf$. We have $\bar{p}(\mathbf{x}) = j\delta$ iff $\tilde{p}(\mathbf{x}) \in B_j$, so that $|\tilde{p}(\mathbf{x}) - j\delta| \leq \delta$ holds conditioned on this event. So

$$\mathsf{sMCE}_{\mathcal{D}}(\bar{p}, \mathcal{C}) = \sum_{j \in [m]} \Pr[\bar{p}(\mathbf{x}) = j\delta] \max_{c \in \mathcal{C}} \left| \mathop{\mathbf{E}}_{\mathcal{D}} [c(\mathbf{x})(\mathbf{y} - j\delta)|\bar{p}(\mathbf{x}) = j\delta] \right|$$

$$= \sum_{j \in [m]} \Pr[\tilde{p}(\mathbf{x}) \in B_j] \max_{c \in \mathcal{C}} \left| \mathop{\mathbf{E}}_{\mathcal{D}} [c(\mathbf{x})(\mathbf{y} - j\delta)|\tilde{p}(\mathbf{x}) \in B_j] \right|$$

$$\leq \sum_{j \in [m]} \Pr[\tilde{p}(\mathbf{x}) \in B_j] \left( \delta + \max_{c \in \mathcal{C}} \left| \mathop{\mathbf{E}}_{\mathcal{D}} [c(\mathbf{x})(\mathbf{y} - \tilde{p}(\mathbf{x}))|\tilde{p}(\mathbf{x}) \in B_j] \right| \right)$$

$$\leq \delta + \sum_{j \in [m]} \Pr[\tilde{p}(\mathbf{x}) \in B_j] \max_{c \in \mathcal{C}} \left| \mathop{\mathbf{E}}_{\mathcal{D}} [c(\mathbf{x})(\mathbf{y} - \tilde{p}(\mathbf{x}))|\tilde{p}(\mathbf{x}) \in B_j] \right| \qquad (25)$$

Let us fix a bucket $B_j$ and a particular $c \in \mathcal{C}$. For $\beta \geq \alpha$ to be specified later we have

$$|\mathbf{E}[c(\mathbf{x})(\mathbf{y} - \tilde{p}(\mathbf{x}))|\tilde{p}(\mathbf{x}) \in B_j]| \leq \Pr[c(\mathbf{x})(\mathbf{y} - \tilde{p}(\mathbf{x})) \geq \beta|\tilde{p}(\mathbf{x}) \in B_j] + \beta \Pr[c(\mathbf{x})(\mathbf{y} - f(x)) \leq \beta|\tilde{p}(\mathbf{x}) \in B_j]$$

$$\leq \frac{\Pr[\tilde{p}(\mathbf{x}) \in \mathrm{Bad}_\beta(c, f) \cap B_j]}{\Pr[\tilde{p}(\mathbf{x}) \in B_j]} + \beta$$

$$\leq \frac{\Pr[\tilde{p}(\mathbf{x}) \in \mathrm{Bad}_\beta(c, f)]}{\Pr[\tilde{p}(\mathbf{x}) \in B_j]} + \beta$$

$$\leq \frac{\alpha/\beta}{\Pr[\tilde{p}(\mathbf{x}) \in B_j]} + \beta.$$

Since this bound holds for every $c$, it holds for the max over $c \in \mathcal{C}$ conditioned on $\tilde{p}(\mathbf{x}) \in B_j$. Hence

$$\sum_{j \in [m]} \Pr[\tilde{p}(\mathbf{x}) \in B_j] \max_{c \in \mathcal{C}} \left| \mathop{\mathbf{E}}_{\mathcal{D}} [c(\mathbf{x})(\mathbf{y} - \tilde{p}(\mathbf{x}))|\tilde{p}(\mathbf{x}) \in B_j] \right| \leq \sum_{j \in [m]} \Pr[\tilde{p}(\mathbf{x}) \in B_j] \left( \frac{\alpha/\beta}{\Pr[\tilde{p}(\mathbf{x}) \in B_j]} + \beta \right)$$

$$\leq \frac{\alpha}{\beta\delta} + \beta,$$

where we use $m = 1/\delta$. Plugging this back into Equation (25) gives

$$\mathsf{sMCE}_{\mathcal{D}}(\bar{p}, \mathcal{C}) = \mathop{\mathbf{E}}_{\mathbf{v} \sim \bar{p}_{\mathcal{D}}} \left[ \max_{c \in \mathcal{C}} \left| \mathop{\mathbf{E}}_{\mathcal{D}|_{\mathbf{v}}} [c(\mathbf{x})(\mathbf{y} - \mathbf{v})] \right| \right] \leq \frac{\alpha}{\beta\delta} + \beta + \delta.$$

Taking $\beta = \sqrt{\alpha/\delta}$ gives the desired claim. ∎

## C.2 Omitted Proofs from Main Result

*Proof of Lemma 3.2.* We will show that for $p, p' \in [0, 1]$ and $t_0 \in I_\ell$, we have

$$\ell(p, t_0) - \ell(p', t_0) \leq |p - p'|B.$$

By the definition of $\ell(p, t)$, we have

$$\ell(p, t_0) - \ell(p', t_0) = (p - p')\ell(0, t_0) + (1 - p - 1 + p')\ell(1, t_0)$$

$$= (p - p')(\ell(0, t_0) - \ell(1, t_0))$$

Taking absolute values and using the Boundedness property gives the desired claim. ∎

*Proof of Claim 3.4.* Suppose that $h \in \text{Lin}(\mathcal{C}, W)$ of the form $h(x) = \sum_{c \in \mathcal{C}} w_c \cdot c(x)$. From Claim 2.7, we know that the multicalibration violation for $c \in \mathcal{C}$ is bounded by $\alpha(v)$ for every $v \in \text{Im}(\tilde{p})$.

$$
\begin{aligned}
|\mathbf{E}[h(\mathbf{x})(\mathbf{y} - v) \mid \tilde{p}(\mathbf{x}) = v]| &= \left| \mathbf{E}\left[ \sum_{c \in \mathcal{C}} w_c \cdot c(\mathbf{x})(\mathbf{y} - v) \mid \tilde{p}(\mathbf{x}) = v \right] \right| \\
&\leq \left( \sum_{c \in \mathcal{C}} |w_c| \right) \cdot \max_{c \in \mathcal{C}} |\mathbf{E}[c(\mathbf{x})(\mathbf{y} - v) \mid \tilde{p}(\mathbf{x}) = v]| \\
&\leq W \cdot \alpha(v)
\end{aligned}
$$

The inequalities follow by Holder's inequality and the assumed bound on the weight of $W$ for $h \in \text{Lin}(\mathcal{C}, W)$. ∎

*Proof of Claim 3.5.* Recall that $\text{Cov}[\mathbf{y}, \mathbf{z}] = \mathbf{E}[\mathbf{yz}] - \mathbf{E}[\mathbf{y}]\,\mathbf{E}[\mathbf{z}]$. For any $h \in \text{Lin}(\mathcal{C}, W)$ we have

$$
\begin{aligned}
|\text{Cov}[\mathbf{y}, h(\mathbf{x})|\tilde{p}(\mathbf{x}) = v]| &= |\mathbf{E}[h(\mathbf{x})(\mathbf{y} - \mathbf{E}[\mathbf{y}])|\tilde{p}(\mathbf{x}) = v]| \\
&= |\mathbf{E}[h(\mathbf{x})(\mathbf{y} - v)|\tilde{p}(x) = v]| + |\mathbf{E}[(v - \mathbf{y})|\tilde{p}(\mathbf{x}) = v]| \\
&\leq (W + 1)\alpha(v)
\end{aligned}
$$

where we use the fact that $h \in \text{Lin}(\mathcal{C}, W)$ and $1 \in \mathcal{C}$. Since $\mathbf{y} \in \{0, 1\}$, this implies the claimed bounds by standard properties of covariance (see [15, Corollary 5.1]). ∎

*Proof of Lemma 3.6.* For any $y \in \{0, 1\}$,

$$
\begin{aligned}
\mathop{\mathbf{E}}_{\mathcal{D}|_v} [\ell(\mathbf{y}, h(\mathbf{x}))|(\tilde{p}(\mathbf{x}), \mathbf{y}) = (v, y)] &= \mathop{\mathbf{E}}_{\mathcal{D}|_v} [\ell(y, h(\mathbf{x}))|(\tilde{p}(\mathbf{x}), \mathbf{y}) = (v, y)] \\
&\geq \ell(y, \mathbf{E}[h(\mathbf{x})|(\tilde{p}(\mathbf{x}), \mathbf{y}) = (v, y)]) && (26) \\
&= \ell(y, \mu(h : v, y)) \\
&\geq \ell(y, \Pi_\ell(\mu(h : v, y))). && (27)
\end{aligned}
$$

where Equation (26) uses Jensen's inequality, and Equation (27) uses the optimality of projection for nice loss functions. Further, by the 1-Lipschitzness of $\ell$ on $I_\ell$, and of $\Pi_\ell$ on $\mathbb{R}$

$$
\begin{aligned}
\ell(y, \Pi_\ell(\mu(h : v, y))) - \ell(y, \Pi_\ell(\mu(h : v))) &\leq |\Pi_\ell(\mu(h : v, y)) - \Pi_\ell(\mu(h : v))| \\
&\leq |\mu(h : v, y) - \mu(h : v)| && (28)
\end{aligned}
$$

Hence we have

$$
\begin{aligned}
&\mathop{\mathbf{E}}_{\mathcal{D}|_v} [\ell(\mathbf{y}, \Pi_\ell(\mu(h : v)))] - \mathop{\mathbf{E}}_{\mathcal{D}|_v} [\ell(\mathbf{y}, h(\mathbf{x}))] \\
&= \sum_{y \in \{0,1\}} \Pr[\mathbf{y} = y|\tilde{p}(\mathbf{x}) = v] \left( \ell(y, \Pi_\ell(\mu(h : v))) - \mathbf{E}[\ell(y, h(\mathbf{x}))|(\tilde{p}(\mathbf{x}), \mathbf{y}) = (v, y)] \right) \\
&\leq \sum_{y \in \{0,1\}} \Pr[\mathbf{y} = y|\tilde{p}(\mathbf{x}) = v] \left( \ell(y, \Pi_\ell(\mu(h : v))) - \ell(y, \Pi_\ell(\mu(h : v, y))) \right) && \text{(By Equation (27))} \\
&\leq \sum_{y \in \{0,1\}} \Pr[\mathbf{y} = y|\tilde{p}(\mathbf{x}) = v] \, |\mu(h : v, y) - \mu(h : v)| && \text{(by Equation (28))} \\
&\leq 2(W + 1)\alpha(v). && \text{(By Equation (9))}
\end{aligned}
$$
∎

# D  Details on Algorithm

Here, we give a high-level overview of the MCBoost algorithm of [20] and weak agnostic learning.

**Definition D.1** (Weak agnostic learning). *Suppose $\mathcal{D}$ is a data distribution supported on $\mathcal{X} \times [-1, 1]$. For a hypothesis class $\mathcal{C}$, a weak agnostic learner* WAL *solves the following promise problem: for some accuracy parameter $\alpha > 0$, if there exists some $c \in \mathcal{C}$ such that*

$$
\mathop{\mathbf{E}}_{(\mathbf{x}, \mathbf{z}) \sim \mathcal{D}} [c(\mathbf{x}) \cdot \mathbf{z}] \geq \alpha
$$

*then* $\mathrm{WAL}_\alpha$ *returns some* $h : \mathcal{X} \to \mathbb{R}$ *such that*

$$\mathop{\mathbf{E}}_{(\mathbf{x},\mathbf{z})\sim\mathcal{D}}[h(\mathbf{x}) \cdot \mathbf{z}] \geq \mathrm{poly}(\alpha).$$

For the sake of this presentation, we are informal about the polynomial factor in the guarantee of the weak agnostic learner. The smaller the exponent, the stronger the learning guarantee (i.e., we want $\mathrm{WAL}_\alpha$ to return a hypothesis with correlation with $\mathbf{z}$ as close to $\Omega(\alpha)$ as possible). Standard arguments based on VC-dimension demonstrate that weak agnostic learning is statistically efficient.

### D.1 MCBoost

The work introducing multicalibration [20] gives a boosting-style algorithm for learning multicalibrated predictors that has come to be known as MCBoost. The algorithm is an iterative procedure: starting with a trivial predictor, the MCBoost searches for a supported value $v \in \mathrm{Im}(\tilde{p})$ and "subgroup" $c_v \in \mathcal{C}$ that violate the multicalibration condition. Note that some care has to be taken to ensure that the predictor $\tilde{p}$ stays supported on finitely many values, and that each of these values maintains significant measure in the data distribution $\mathcal{D}_{\tilde{p}}$. In this pseudocode, we ignore these issues; [20] handles them in full detail.

Importantly, the search over $\mathcal{C}$ for condition (29) can be reduced to weak agnostic learning. Intuitively, we pass WAL samples drawn from the data distribution, but labeled according to $\mathbf{z} = \mathbf{y} - v$ when $\tilde{p}(\mathbf{x}) = v$.

*Lemma 3.8.* The iteration complexity of MCBoost is directly (inverse quadratically) related to the size of the multicalibration violations we discover in (29). A standard potential argument can be found in [20].

By the termination condition, we can see that $\tilde{p}$ must actually be $(\mathcal{C}, \alpha)$-swap multicalibrated. In particular, when the algorithm terminates, then for all $v \in \mathrm{Im}(\tilde{p})$, we have that

$$\max_{c_v \in \mathcal{C}} \mathbf{E}[c_v(\mathbf{x}) \cdot (\mathbf{y} - v) \mid \tilde{p}(\mathbf{x}) = v] \leq \mathrm{poly}(\alpha) \leq \alpha.$$

Therefore, averaging over $\mathbf{v} \sim \mathcal{D}_{\tilde{p}}$, we obtain the guarantee. ∎

*Corollary 3.9.* By Lemma 3.8, we know that $\tilde{p}$ returned by MCBoost is $(\mathcal{C}, \alpha)$-swap multicalibrated. By Theorem 3.3, $\tilde{p}$ is equivalently a $(\mathcal{L}_{\mathrm{cvx}}, \mathcal{C}, \alpha')$-swap omnipredictor for some polynomially-related $\alpha'$. In other words, by Claim 2.4, if we post-process $\tilde{p}$ according to $k_\ell$ for any nice convex loss function $\ell$, we obtain an $(\ell, \mathcal{C}, \varepsilon)$-swap agnostic learner. Taking $\alpha = \mathrm{poly}(\varepsilon)$ sufficiently small, we obtain the swap agnostic learning guarantee. ∎

---

**Algorithm 2** MCBoost

---

**Parameters:** hypothesis class $\mathcal{C}$ and $\alpha > 0$
**Given:** Dataset $S$ sampled from $\mathcal{D}$
**Initialize:** $\tilde{p}(x) \leftarrow 1/2$.
**Repeat:**
if $\exists v \in \mathrm{Im}(\tilde{p})$ and $c_v \in \mathcal{C}$ such that

$$\mathbf{E}[c_v(\mathbf{x}) \cdot (\mathbf{y} - v) \mid \tilde{p}(\mathbf{x}) = v] > \mathrm{poly}(\alpha) \qquad (29)$$

update $\tilde{p}(x) \leftarrow \tilde{p}(x) + \eta c_v(x) \cdot \mathbf{1}[\tilde{p}(x) = v]$
**Return:** $\tilde{p}$

---

