# OpenReview forum: "Swap Agnostic Learning, or Characterizing Omniprediction via Multicalibration"
_NeurIPS.cc/2023/Conference — NeurIPS 2023 poster_

### Official Review · Reviewer_Xn6q · 2023-06-28

**Soundness:** 4 excellent
**Presentation:** 3 good
**Contribution:** 3 good
**Rating:** 7
**Confidence:** 4

**Summary:**

This paper introduces and studies 3 learning theoretic problems, namely "swap" versions of agnostic learning, omniprediction, and multicalibration. These swap versions are inspired by the notion of "swap regret" in online learning. They show that in some settings, all three of these learning-theoretic problems are equivalent (up to polynomial factors). Then they also introduce another notion called Swap Loss Outcome Indistinguishability and study relationships between it and the others.

**Strengths:**

- The paper is clearly written, and all the definitions of the different learning problems make sense. The proofs are easy to follow/digest.
- The results seem novel, and provide a relatively complete picture of the relationships between the different learning problems that are introduced.

**Weaknesses:**

- The main question which is not really addressed is why one should care about these learning problems? Agnostic learning and multicalibration are important problems because they have tremendous practical significance. Maybe the authors can touch upon this in the rebuttal. At first glance (since I do not have much background in the field of multicalibration), it seems like the authors introduced three "brand new" learning problems and proved relationships between them.

**Questions:**

1. Could the authors address why one should study these learning problems (swap versions)?
2. While the paper was motivated by studying omniprediction and multicalibration, the results herein focus on the swap version. What are the high level implications for the relationships between omniprediction and multicalibration?
3. Can one do an online2batch reduction for the swap versions of online learning and agnostic learning?
4. How do the results depend on the size of the support of $\tilde{p}$? Do they become less interesting if $\tilde{p}$ is assumed to be a continuous-valued predictor (as is done in many practical scenarios)?
5. Could the authors state some examples of "nice" loss functions? Are all the loss functions commonly used in learning theory, e.g, square loss, hinge loss, exp loss, log loss, etc all "nice"?

Minor typos/clarifications:
- line 63: "defintions"
- I may have missed this, but what is meant by the notation $\ell_2$ and $\ell_4$? In some cases, it seems like it means square loss, but in others it is used to denote a loss function that is picked for a value $v$.
- What is meant by "sparse"? Typically it means a bound on $\ell_0$ (or the number of nonzero weights), but it seems like you are using it to mean $\ell_1$ bound.
- line 297: "ot"
- line 188: do you mean $\lVert \mathcal{C} \rVert_\infty \le 1$?

**Limitations:**

Yes, n/a.

---

> ### Author Rebuttal · Authors · 2023-08-07
>
> We thank the reviewer for the questions.  We hope the main response above helps to answer the motivation for our study.  These swap variants are natural in a few ways.  Perhaps the way to motivate these notions as natural is through multicalibration.  Multicalibration was introduced as a notion of algorithmic fairness in prediction tasks.  It makes no reference to loss functions or to swapping.  And yet, we show that this notion is tightly characterized by the notion of Swap Agnostic Learning.  Even though initially, it seems like a simple syntactic extension of the Agnostic Learning paradigm, it is clear from our initial study that swap agnostic learning may have a deep and rich theory to uncover.
>
> Hopefully questions 1. and 2. are addressed in the main response.
>
> 3.  Obtaining sample-efficiency was a key consideration in the original work of [20], but they do not use an online to batch conversion.  It is a very interesting question for future investigation.  The online version of the problem has only very recently been studied (in a follow-up to our work).
>
> 4.  Restricting ourselves to a finite support size is a technical convenience.  As we show, by rounding, one can achieve approximate versions of all these notions.
>
> 5.  All of the loss functions you mention are nice! ( Page 10 of the original Omnipredictors paper (GKRSW’22) shows this explicitly.)  In fact, it’s a bit of an exercise in creativity to think up loss functions that are not nice.  Thank you for flagging this.  We will add a sentence about this fact, which will no doubt improve the clarity of the presentation.

---

> > ### Comment · Reviewer_Xn6q · 2023-08-14
> > **Thanks**
> >
> > Thank you for clarifying. I have adjusted my score.

---

### Official Review · Reviewer_if87 · 2023-07-05

**Soundness:** 4 excellent
**Presentation:** 4 excellent
**Contribution:** 3 good
**Rating:** 8
**Confidence:** 4

**Summary:**

The authors propose a stronger variant of agnostic learning called *swap* agnostic learning, and show that this notion of learning is equivalent to “swap” variants of multicalibration and omniprediction. Multicalibration means that the predictor is simultaneously calibrated on a given collection of subpopulations $\mathcal{C}$. Omniprediction means that a *single* predictor’s output values can be post-processed so that its performance is competitive with the best from a given hypothesis class $\mathcal{C}$ w.r.t. multiple loss functions. Here, the post-processing must depend only on the given loss function and *not* the covariates.

Multicalibration is related to algorithmic fairness since it means that “chance” predictions of a given (potentially malicious) predictor “mean the same thing” across many subpopulations. Omniprediction is useful for training machine learning (ML) models since it means that one can train a *single* predictor that performs well w.r.t. multiple loss functions. Thus, an omnipredictor obviates the need to train separate models for each loss function.

Previous work has shown that multicalibration is sufficient for omniprediction, thereby showing that satisfying fairness constraints may lead to practical benefits in machine learning. The question of whether multicalibration is *necessary* for omniprediction is still open. This work takes a step towards this question by showing that their *swap* analogues are equivalent to each other.

**Strengths:**

- **Contribution.** The paper makes a significant and timely contribution to the fast-growing literature on multicalibration by showing that seemingly distinct notions of success in learning and algorithmic fairness are in fact equivalent. Key to their result is their definition of swap agnostic learning, a stronger learning requirement than (vanilla) agnostic learning, and the swap variants of multicalibration and omniprediction. The swap analogues of each notion enable tight connections among themselves.
- **Exposition.** The paper is overall well-written. Ideas are presented clearly and concisely. The authors also do a good job situating their results in relation to previous work.

**Weaknesses:**

**Motivation behind swap multicalibration and swap omniprediction.** As a stand alone concept, i.e., for purposes other than establishing equivalence between the notions, the motivation behind the definition of swap multicalibration and swap omniprediction is unclear to me. I expand on this point in the Questions section.

**Questions:**

- Are there provable separations between swap and vanilla versions of each notion (multicalibration, omniprediction)? What separations are known and which ones are unknown?
- Multicalibration can be interpreted as “calibration w.r.t. any subpopulation in $\mathcal{C}$”.  Does swap multicalibration have such simple interpretation? How would one motivate swap multicalibration as a stand alone concept? Ditto for swap omniprediction.
- Calibration has a long history dating back to early research on weather forecasts and their assessments. The natural setting of this early literature is sequential prediction (i.e., inputs are *adversarial* sequences consisting only of labels $y$ rather than i.i.d. samples $(x,y)$ with covariates $x$ drawn from an unknown, but fixed, distribution). Even in the sequential setting, multicalibration has a natural interpretation: calibration w.r.t. multiple subsequence selection rules (see e.g., [Daw85]). How should one think of infinitary analogues of swap mulitcalibration?

**Minor editorial comments**
- In the abstract, it’s not clear what $p$ refers to when the authors say “the predictor wins if $p$ competes with the adaptive adversary’s loss”. Did the authors mean $h$?
- The projection operator $\Pi_\ell$ in Definition 3.1 is not defined anywhere in the paper.
- In Theorem 4.2, the word “incomparable” is too vague and uninformative, even for an informal theorem statement. I would suggest the authors provide a more concrete statement.

**References**
- [Daw85]: Dawid, A. P. Calibration-based empirical probability. *Annals of Statistics*. 1985

**Limitations:**

Yes.

---

> ### Author Rebuttal · Authors · 2023-08-07
>
> To reiterate from the main response, multicalibration and swap multicalibration are essentially the same, so there is little additional motivation.  For swap omniprediction (and similarly agnostic learning), the guarantee seems to be much stronger.  In some sense, one can view this as pushing the limits of supervised learning:  how strong can we make the adversary that the learner is up against, and still be able to achieve the solution concept?  We show that despite the very strong framing, swap omniprediction can be achieved through weak agnostic learning.
>
> Re:  How should one think of infinitary analogues of swap mulitcalibration?
> Fantastic research question!  Very recent work (posted to the arXiv) looks at exactly this question, particularly in the online setting.

---

> > ### Comment · Reviewer_if87 · 2023-08-14
> >
> > Thank you for your response!
> >
> > P.S. This is probably irrelevant to the current rebuttal/discussion phase, but it would be great if the authors could provide a reference for the recent arXiv paper on the infinite sequence setting.

---

### Official Review · Reviewer_HdZq · 2023-07-07

**Soundness:** 3 good
**Presentation:** 2 fair
**Contribution:** 3 good
**Rating:** 5
**Confidence:** 2

**Summary:**

The authors study swap agnostic learning, or a game between a predictor and an adversary where the predictor has to come up with a family of hypotheses to compete with a strong adversary's loss. They prove a surprising feasibility result for swap agnostic learning for any convex loss by proving that swap agnostic learning is equivalent to swap variants of omniprediction and multicalibration. Furthermore, they connect these ideas even further by relating them under the umbrella of outcome indistinguishability.

**Strengths:**

+ The equivalence result seems like it is a novel and useful finding -- however, I was not familiar with the notions of omniprediction and multicalibration.
+ The technical contributions of this paper are insightful and nontrivial.
+ This paper connects results from different areas of ML, which is always nice to see.

**Weaknesses:**

- I found the paper to be rather inaccessible in its lack of intuition about results and its overeagerness to use jargon. I acknowledge that this is not an area I am an expert in, but I found it rather hard to follow.


**Questions:**

None

**Limitations:**

Yes

---

> ### Author Rebuttal · Authors · 2023-08-07
>
> We appreciate you bringing up these points on presentation.  If there is any jargon that you feel is particularly inaccessible to the general machine learning audience, we are happy to incorporate definitions and clarifications.  We will be cognizant of aiming for accessibility in our revisions for the camera-ready version, which no doubt, will improve the paper.  Thanks!

---

### Official Review · Reviewer_XpDf · 2023-07-09

**Soundness:** 3 good
**Presentation:** 3 good
**Contribution:** 3 good
**Rating:** 8
**Confidence:** 3

**Summary:**

The paper introduces the notion of swap agnostic learning, and proves the feasibility of swap agnostic learning for any convex loss using only weak agnostic learners. To prove this result, the authors establish the equivalence between swap agnostic learning and the swap variants of multi-calibration and omniprediction, and show that the existing multicalibration algorithm of [20] already guarantees the stronger notion of swap multicalibration. They also show that the notion of swap loss outcome indistringuishability captures all existing notions of omniprediction and multicalibration.

**Strengths:**

- The paper explores the important connection between the group-fairness notion of multi-calibration and the loss minimization notions of agnostic learning and omniprediction, and establishes many interesting equivalences under swap variants. This equivalence result provides new insights into what multi-calibration can offer and deeps the understanding of the relevant notions.

- The paper is well-written and clearly explains the problem and its contributions.

**Weaknesses:**

The paper would benefit from additional discussions on the differences between the 'swap' variations and their standard counterparts. For instance, clarifying how 'swap multicalibration' diverges from the conventional concept introduced in [20] would be beneficial. Further, it would be insightful if the paper could elaborate on why equivalences under the 'swap' notions are easier to establish. What are the challenges involved in extending these to the standard definitions? This additional discussion could give readers a more thorough understanding of the subject.

**Questions:**

- In line 5 of the abstract, what does $p$ refer to? Is this supposed to be $h$?
- In Definition 2.5, multi-calibration is defined in terms of the expectation of the absolute difference over $v\sim\cD_{\tilde{p}}$, which contrasts with the definition involving $\max_v$ as in Definition 2.6 from reference [20]. How critical is this choice of definition for your results? If multi-calibration were defined in terms of the maximum rather than the expectation, would this make swap multi-calibration and multi-calibration equivalent? Could this be why existing algorithms for achieving multi-calibration, such as those in references [20,15], are able to guarantee swap multi-calibration without modification?

**Limitations:**

Yes.

---

> ### Author Rebuttal · Authors · 2023-08-07
>
> Swap and standard multicalibration are equivalent in the max version you describe.  Unfortunately, the max version is hard to achieve from sample access alone. This is why all known algorithms work with (some variant of) the expectation version, which allows us to ignore unlikely predictions.  (Even the original formulation of (Hebert-Johnson, et al. ICML 2018) uses this on-average version under the hood.) None of the algorithms mentioned can guarantee the max version, since they all work from samples alone, they essentially guarantee the expectation version.  Still, the algorithms work by searching for a successful auditing function $c \in \mathcal{C}$, conditioned on each prediction value (i.e., the level sets of the current predictor).  This conditioning, followed by searching for $c \in \mathcal{C}$, flips the quantifiers from the standard multicalibration definition, and gives rise to an algorithm that naturally guarantees swap multicalibration.

---

> > ### Comment · Reviewer_XpDf · 2023-08-15
> >
> > Thank you for answering my questions!

---

### Official Review · Reviewer_7A6v · 2023-07-09

**Soundness:** 4 excellent
**Presentation:** 4 excellent
**Contribution:** 3 good
**Rating:** 7
**Confidence:** 4

**Summary:**

This paper studies algorithmic fairness from a learning theoretic lens, specifically extensions of the recent notion omniprediction by Gopalan et al and its connections to to agnostic learning - multicalibration is sufficient to obtain omniprediction and guarantee agnostic learning for all convex loss functions simultaneously. The main contributions are in introducing a new learning paradigm called Swap Agnostic Learning (SAL), and subsequently new notions of "swap" omniprediction and multicalibration in a similar vein, and proving that they are all equivalent. The authors show that the standard multicalibration algorithm of Herbert-Johnson et al. in fact guarantees swap multicalibration algorithm, and consequently can also be used to obtain a SAL guarantee.

The contributions all hinge on the introduction of the new learning task - swap agnostic learning. SAL requires, informally, that a predictor plays a hypothesis h such that on any level set of their predictor - i.e. set of all x's such that h(x) = v for some fixed v - they have no incentive to change their predictions to any other hypothesis from the hypothesis class on that set. This contrasts the standard agnostic learning which says that the predictor is competitive with the same, single error-minimizing hypothesis in expectation over all the level sets.

The swap notion of omniprediction is similar. A standard omnipredictor for a class of losses L and hypothesis C is a predictor that can be post-processed (in a way that depends on the chosen loss from L) such that it is competitive with the best hypothesis for each loss in L simultaneously. The swap variant requires that the omnipredictor be competitive (up to a level-set dependent post-processing) the best hypothesis on each level set for any assignment of loss functions to level sets.

Section 3 proceeds with the proof of equivalence between these three notions and a brief explanation of how the standard McBoost multiwcalibration algorithm in fact satisfies swap multicalibration, and consequently can be used to achieve the other two swap notions.

**Strengths:**

This paper makes a significant contribution in the area of omniprediction and multicalibration, which are currently popular topics in the study of algorithmic fairness. The newly introduced notion of SAL is interesting and provides a complete picture of the connections between the swap variants of mutlicalibration, omniprediction, and agnostic learning.

Though the content is technically and notationally dense, the presentation is clear.

**Weaknesses:**

A minor suggestion is that if possible, adding a bit of exposition to the swap omniprediction definition to guide the reader through all the moving parts would be helpful. It feels a little trickier to parse than the swap multi calibration and swap agnostic learning definitions. Also, per the below question, maybe adding some exposition describing why these swap variants are natural / intuitive could be helpful.

Simiarly, in Section 3.1, if possible, proof sketches/intuition would be useful as well. There are maybe one or two parts where it is done but the section feels largely like a proof with limited expositional guidance.

**Questions:**

I understand the technical conditions of swap AL, MC, omniprediction and how they are different from the standard notions, but I am slightly confused by the motivation. The original notions seem natural in some sense, is there a reason why these additional technical conditions for the swap variants are intuitive or useful - e.g. giving adversary the additional control down to the granularity of level sets? If I understand correctly, the analogous versions of connections established in this work for all of the "non-swap" notions of these learning tasks are already shown, so again I am slightly confused what the advantage of these swap variants are.

**Limitations:**

-

---

> ### Author Rebuttal · Authors · 2023-08-07
>
> To be very clear, the equivalences we show for the swap variants are NOT known to be true in the standard models:  the implications are only known to go in one direction.  In fact, our work suggests that they do not generally hold.  This is a key contribution of our work.  On a technical note, (Gopalan et al., ITCS 2023) demonstrate some equivalences in the standard learning model, but they cannot give guarantees for convex omniprediction, which we study here.

---

> > ### Comment · Reviewer_7A6v · 2023-08-16
> >
> > Thanks!

---

### Official Review · Reviewer_7mxb · 2023-07-26

**Soundness:** 2 fair
**Presentation:** 3 good
**Contribution:** 2 fair
**Rating:** 5
**Confidence:** 2

**Summary:**

This paper provides a new learning framework referred to as Swap Agnostic Learning, a variant of agnostic learning aiming to a stronger goal.
It is shown that this framework is equivalent ot swap variants of the notions of Omniprediction and Multicalibration.
An algorithm for Swap Agnostic Learning is provided, which is shown to work well for any convex losses.

**Strengths:**

- A new problem setup is introduced along with its interpretation, which would be of interest for readers in the NearIPS community.
- The relationship with various existing studies is well explained.

**Weaknesses:**

There are unclear points in the algorithm procedure and analysis.
For example, the paper states that MCBoost from [20] is used, but it is not clear what the corresponding algorithm in the paper [20] is.
Algorithm 2 given in Appendix B.1 looks very different from algorithms in [20].
In Algorithm 2, there is no explanation of how to set the parameter $\eta$.
I cannot find a proof that the iterations in Algorithm 2 terminate.

In the algorithm section, the fact that the MCBoost algorithm, originally designed to guarantee multicalibration, actually also guarantees swap multicalibration, would be a very important and nontrivial point, in my understanding.
I believe it would be better if a more detailed explanation of this fact and the intuition were added to the text.

**Questions:**

I would appreciate an answer to the concerns in "Weaknesses".

**Limitations:**

I have no concerns about the limitations and potential negative societal impact.

---

> ### Author Rebuttal · Authors · 2023-08-07
>
> We apologize for some of the confusions.  Multicalibration is a fairly young notion, but has had significant developments and contributions to the literature in the past few years.  As such, the presentation and nomenclature has become a bit ambiguous.  Let us clarify.
>
> MCBoost—i.e., Mutlicalibration Boosting—is a name that has come to refer to Algorithm 3.2 of [20] over the years.  While [20] presents the algorithm in terms of sets (rather than hypotheses) and only incorporates the reduction to weak agnostic learning in Section 4, the literature has settled on referring to this general learning paradigm as MCBoost, and attributing it to [20].
>
> MCBoost operates iterative, in a boosting-style fashion.  Importantly, in each iteration, the input domain is partitioned by the current predictions.  This split of the domain effectively conditions on the level sets of the current predictor, then searches over $c \in \mathcal{C}$.  In other words, even though the algorithm was designed to satisfy standard multicalibration, the order of quantifiers matches that of swap multicalibration.  This is why MCBoost actually guarantees the stronger variant of swap multicalibration.
>
> Thanks for bringing up these points!  We are happy to clarify these points in our presentation of MCBoost.  We hope that incorporating these changes would improve your impression of the manuscript.

---

> > ### Comment · Reviewer_7mxb · 2023-08-16
> >
> > Thank you for the response. My question in the review has been resolved. I have no additional questions at this time.

---

### Author Rebuttal · Authors · 2023-08-07

We thank all of the reviewers for their thoughtful, and overall, positive feedback on the submission on Swap Agnostic Learning, Multicalibration, and Omniprediction.  We hope to clarify our views on the distinction between the Swap and Standard learning tasks—a point that came up across many reviews—and the motivation for studying the swap variants.

Our motivation for introducing Swap Agnostic Learning comes from trying to understand the relationship between (standard) multicalibration and omniprediction.  Prior work shows that multicalibration implies (convex) omnipredction, but leaves open the question of whether omniprediction implies multicalibration.  The present work sheds new light on this question, effectively answering it in the negative:  standard omniprediction does NOT seem to imply multicalibration.  The answer to this question comes by introducing the idea of Swap Learning, in the agnostic learning / omniprediction setting, as well as in multicalibration.

Importantly, for multicalibration, Swap multicalibration and Standard multicalibration are essentially the same notion!  Claim 2.9, as well as our analysis of prior algorithms for multicalibration, shows that standard multicalibration is already strong enough to capture the swap variant.  We will emphasize this point in the future presentation.  The distinction (definitionally and algorithmically) between swap and standard multicalibration is minimal:  our results suggest that one should really think of them as a single notion of multicalibration.

For agnostic learning and omniprediction, however, Swap Learning is much stronger than Standard Learning.  We prove separations between swap and standard omniprediction, but due to space requirements, the formal statements and proofs were deferred to Appendix C.  The separations are summarized in Appendix Figure 1.  We are happy to elaborate on these separations in the main body for the camera-ready version.

Our main result shows that (Swap) Multicalibration, Swap Agnostic Learning for $\ell_2$, and Swap Convex Omniprediction are all equivalent.  By this equivalence alone, Swap Agnostic Learning is a well-motivated learning model—it is an alternative view on a popular notion of algorithmic fairness in prediction.  Further, by the separation of Swap Convex Omniprediction and Standard Convex Omniprediction (proved in the appendix), we effectively separate Multicalibration from Omniprediction:  multicalibration suffices for omniprediction, but the converse should not generally be true.

Thus, in the standard setting, the state of the art is that multicalibration suffices for (convex) omniprediction.  We leave open the fantastic question of what (if any) notion of multi-group fairness characterizes (i.e., is equivalent to) standard omniprediction.  (Gopalan et al., ITCS 2023) make some progress on this question for non-convex losses, which may provide a roadmap to this question for convex losses.

The reviewers' comments and questions are very helpful, in forcing us to articulate these points more clearly!  In the camera-ready version, we will include prose (as above) to better situate our study of the swap variants of learning.

---

### Decision · Program_Chairs · 2023-09-21

**Decision:**

Accept (poster)

**Comment:**

The paper introduces and explores the concept of “swap” versions of agnostic learning, omniprediction, and multicalibration, establishing their equivalence under certain conditions. While reviewers initially sought clarification on the motivation and practical relevance of these variants, the authors effectively addressed these concerns by emphasizing their significance in bridging the gap between learning theory and fairness considerations. Overall, the paper is technically sound and makes valuable contributions, and with further refinements and clarifications in the final version, it has the potential to stimulate further research in this area.